# Active Regression for Single-Index Models with Unknown Link Functions

**Chansophea Wathanak In** [1] **Yi Li** [1 2] **Wai Ming Tai** [3] **Xuan Wu** [4]

## Abstract

This paper studies active regression for single-index models under general $\ell_p$-loss with an unknown 1-Lipschitz link function $f$, formulated as $\min_{f,x} \|f(Ax) - b\|_p^p$ with full access to $A$ but coordinate-query access to $b$. Prior work established upper bounds for known link functions for all $p \geq 1$ and for unknown link functions only in the $p = 2$ case, together with lower bounds for $p \leq 2$. This work addresses the more challenging setting of unknown link functions and general $p \geq 1$. A non-adaptive sampling algorithm is presented that achieves a $(1+\epsilon)$-approximation using $O(d^{p/2\vee 1}/\epsilon^{p\vee 2} \operatorname{poly}\log(n/\epsilon))$ queries. Nearly tight lower bounds are also established for non-adaptive queries when $p > 2$. These results close much of the remaining gap in active $\ell_p$ regression for single-index models.

## 1. Introduction

The $\ell_p$-regression problem is a fundamental task in randomized numerical linear algebra (RandNLA). Given a data matrix $A \in \mathbb{R}^{n \times d}$ with $n \gg d$ and a label vector $b \in \mathbb{R}^n$, the goal is to find a vector $x$ that minimizes the residual $\|Ax - b\|_p$. A standard paradigm to reduce the computational complexity is row sampling: one constructs a (randomized) row-sampling matrix $S$, meaning that each row has exactly one nonzero entry, such that the solution $\hat{x}$ to the sketched problem, i.e. $\hat{x} = \arg\min_x \|S(Ax - b)\|_p$, approximately solves the original problem in the sense that

$$\|A\hat{x} - b\|_p \leq (1 + \epsilon) \min_x \|Ax - b\|_p$$

with probability at least 0.9. After over a decade of intensive research, the sampling complexity for $\ell_p$-regression is now

---

[1]School of Physical and Mathematical Sciences, Nanyang Technological University, Singapore [2]College of Computing and Data Sciences, Nanyang Technological University, Singapore [3]Independent Researcher [4]John Hopcroft Center for Computer Science, Shanghai Jiaotong University, China. Correspondence to: Yi Li <yili@ntu.edu.sg>.

*Proceedings of the 43rd International Conference on Machine Learning*, Seoul, South Korea. PMLR 306, 2026. Copyright 2026 by the author(s).

well-understood for all values of $p \geq 1$; see, e.g., (Woodruff, 2014; Yasuda, 2024). In particular, $S$ can be constructed by sampling rows of $(A\ b)$ according to the Lewis weights (a generalization of leverage scores) (Ledoux & Talagrand, 1991; Cohen & Peng, 2015). The resulting sample complexity is $\tilde{O}(d/\epsilon^2)$ rows for $p = 1$ and $\tilde{O}(d^{\max\{p/2,1\}}/\epsilon)$ rows for $p > 1$ (Yasuda, 2024).

Remarkably, this framework extends to *active regression*, where the matrix $A$ is fully known but access to $b$ is restricted to coordinate queries. In this setting, one can form $S$ using only Lewis weights of $A$. The sample complexity to achieve the same $(1 + \epsilon)$-approximation guarantee is $\tilde{O}(d/\epsilon^2)$ queries for $p = 1$ (Parulekar et al., 2021) and $\tilde{O}(d^{\max\{p/2,1\}}/\epsilon^{p-1})$ queries for $p > 1$ (Musco et al., 2022). Both bounds are known to be tight up to logarithmic factors, even when the samples are allowed to be adaptive (Parulekar et al., 2021; Yasuda, 2024).

Despite the mature understanding of linear $\ell_p$-regressions, linear models cannot capture common nonlinear behaviours, such as piecewise-linear activations (e.g. ReLU) arising from neural architectures. This limitation has motivated a shift to nonlinear regression problems in recent years, notably the *single-index model*, which generalizes linear regression by composing the linear predictor $Ax$ with a link function $f$.

In this paper, we consider Lipschitz link functions and study the single-index problem as a purely deterministic regression task:

$$\min_{f \in \mathsf{Lip}_1, x \in \mathbb{R}^d} \|f(Ax) - b\|_p, \tag{1}$$

where

$$\mathsf{Lip}_L = \{f \in C(\mathbb{R}) : |f(x) - f(y)| \leq L|x - y|$$
$$\text{for all } x, y \in \mathbb{R} \text{ and } f(0) = 0\}$$

and $f$ is applied entrywise, i.e., $f(z) = (f(z_1), \ldots, f(z_n))$ for $z \in \mathbb{R}^n$. The goal is to approximately solve the regression problem in the active setting introduced earlier using as few queries to coordinate of $b$ as possible.

The single-index regression problem was first investigated from a RandNLA perspective by Gajjar et al. (2023) in the case of $p = 2$ with a *known* link function $f$, i.e., the minimization in (1) is only over $x \in \mathbb{R}^d$. With a query complexity of $O(d^2/\epsilon^4)$, their algorithm outputs $\hat{x}$ that satisfies

a constant-factor error bound

$$\|f(A\hat{x}) - b\|_p^p \le C \left( \|f(Ax^*) - b\|_p^p + \epsilon\|Ax^*\|_p^p \right),$$

with probability at least 0.9. Here, $C$ is an absolute constant, $x^*$ denotes an optimal solution. It was also shown in (Gajjar et al., 2023) that the additive error is necessary if one aims for query complexity polynomial in $d$. This complexity was subsequently improved to $\tilde{O}(d/\epsilon^2)$ in (Gajjar et al., 2024) and generalized to $\tilde{O}(d^{\max\{p/2,1\}}/\epsilon^4)$ for $p \ge 1$ in (Huang et al., 2024). The current state of the art is (Li & Tai, 2025), which achieves a query complexity of $\tilde{O}(d^{\max\{p/2,1\}}/\epsilon^{\max\{p,2\}})$ for $p \ge 1$ and a $(1 + \epsilon)$-approximation, i.e. the output $\hat{x}$ satisfies

$$\|f(A\hat{x}) - b\|_p^p \le (1 + \epsilon)\|f(Ax^*) - b\|_p^p + \epsilon\|Ax^*\|_p^p \quad (2)$$

with probability at least 0.9. Moreover, Li & Tai (2025) showed that this query complexity is tight up to logarithmic factors for $1 \le p \le 2$ and that the $1/\epsilon^p$ dependence is tight for $p > 2$, leaving open the question of a matching lower bound for $p > 2$.

Much less is known when the link function is *unknown*, which is exactly (1) and our focus in this paper. To date, the only result in this setting is due to Gajjar et al. (2024), who obtained a query complexity of $O(d/\epsilon^2 \operatorname{poly} \log n)$ and a constant-factor approximation for the special case $p = 2$ only. Their algorithm outputs a pair $(\hat{f}, \hat{x})$ satisfying the guarantee:

$$\|\hat{f}(A\hat{x}) - b\|_p^p \le C \left( \|f^*(Ax^*) - b\|_p^p + \epsilon\|Ax^*\|_p^p \right),$$

where $C$ is an absolute constant and $(f^*, x^*)$ denote the optimal solution to (1). However, their analysis is very specific for $p = 2$ and does not extend to other values of $p$ easily. Consequently, a significant gap remains in the understanding of single-index regression with unknown link functions.

### 1.1. Our Results

Our first contribution is an upper bound for the active single-index regression problem (1), achieving a $(1 + \epsilon)$-approximation for general $p \ge 1$. This result strictly improves (Gajjar et al., 2024) in both the approximation guarantee and the range of $p$. In particular, we show that when the link function $f$ is unknown, the query complexity is only $\log n$ factors larger than the case of known $f$, consistent with known behaviour for the special case $p = 2$.

**Theorem 1.1** (Informal version of Theorem 3.2). *Let $A \in \mathbb{R}^{n \times d}$, $b \in \mathbb{R}^n$ and $\epsilon > 0$. There exists a randomized algorithm which, with probability at least 0.9, makes $O(d^{1 \vee \frac{p}{2}}/\epsilon^{p \vee 2} \cdot \operatorname{poly} \ln n)$ non-adaptive queries to the entries of $b$ and returns $(\hat{f}, \hat{x}) \in \operatorname{Lip}_1 \times \mathbb{R}^d$ satisfying the $(1 + \epsilon)$-approximation guarantee*

$$\|\hat{f}(A\hat{x}) - b\|_p^p \le (1 + \epsilon)\|f^*(Ax^*) - b\|_p^p + \epsilon\|Ax^*\|_p^p. \quad (3)$$

We remark that our algorithm remains valid if, in the problem formulation (1), the link function $f$ is restricted to a subset $\mathcal{F} \subset \operatorname{Lip}_1$ instead of ranging over the entire $\operatorname{Lip}_1$.

Our second contribution partially fills the lower bound gap mentioned earlier. We show that the known upper bound $\tilde{O}(d^{p/2}/\epsilon^p)$ for $p > 2$ in the case of known $f$ is tight up to logarithmic factors for *non-adaptive queries*. This result implies that the standard row-sampling paradigm, as the state-of-the-art algorithm of (Li & Tai, 2025), cannot be substantially improved in this query model.

**Theorem 1.2** (Informal version of Theorem 4.3). *Suppose that $p > 2$ is a constant, $\epsilon > 0$ is sufficiently small, $d$ is at least some constant depending only on $p$. Every randomized algorithm that makes only non-adaptive queries to $b$ and outputs $\hat{x} \in \mathbb{R}^d$ which satisfies (3) with probability at least 0.9 must make $\Omega_p(d^{p/2}/\epsilon^p)$ queries to the entries of $b$.*

### 1.2. Technique Overview

**Upper Bound.** The algorithm is a simple, non-adaptive sampling scheme following the template in (Li & Tai, 2025; Gajjar et al., 2024): We perform weighted sampling of the coordinates of $b$ where the weights are determined by the Lewis weights of $A$. While the algorithm is conceptually simple, the main technical contribution lies in the analysis required to handle the unknown link function.

Our analysis builds on the framework introduced in (Li & Tai, 2025), incorporating the ideas for unknown link functions from (Gajjar et al., 2024). To simplify the exposition, we assume that $A$ has uniformly bounded Lewis weights $O(d/n)$ and let $\psi_1, \ldots, \psi_n$ be i.i.d. Bernoulli variables such that $\mathbb{E}\psi_i = \alpha$. Let $T$ be a bounded region in $\operatorname{Lip}_1 \times \mathbb{R}^d$ and $v$ be a fixed vector in $\mathbb{R}^n$. The key technical task is to control a uniform sampling error of the following form:

$$\Psi = \sup_{(f,x) \in T} \left| \sum_{i=1}^n \left( \frac{1}{\alpha} \psi_i - 1 \right) Z_i(f, x) \right|, \quad (4)$$

where

$$Z_i(f, x) = |(f(Ax) - v)_i|^p - |(f^*(Ax^*) - v)_i|^p$$

denotes the residual difference relative to a fixed reference point $(f^*, x^*) \in \operatorname{Lip}_1 \times T$.

Following the framework of (Li & Tai, 2025), $\Psi$ is bounded by separating coordinates $i$ according to the magnitude of the residual at the reference point $(\bar{f}, \bar{x})$ and the Lewis weights of $A$. Contributions from coordinates with large residuals or negligible Lewis weights can be controlled using arguments similar to those in (Li & Tai, 2025). The core technical challenge is to control the remaining contribution from the set of coordinates $J$ that simultaneously have small residuals and non-negligible Lewis weights. This term, denoted $\Psi'$, is the focus of the subsequent analysis. After

symmetrization, we have

$$\mathbb{E}\,\Psi' \lesssim \mathbb{E}_{\xi,\psi} \frac{1}{\alpha} \sup_{(f,x)\in T} \left| \sum_{i\in I} \xi_i Z_i(f,x) \right|,$$

where $\xi_i$ are independent Rademacher variables and $I \subseteq J$ is the random set of indices $i$ with $\mathbb{E}\,\psi_i = 1$. As in (Gajjar et al., 2024), this term can be controlled via Dudley's integral applied to an Rademacher process indexed by $T$.

A natural approach is to decouple $f$ and $x$, reducing the Dudley's integral for $T$ to two Dudley integrals, one over $\pi_1(T)$ and one over $\pi_2(T)$, where $\pi_1(T)$ and $\pi_2(T)$ denote the projections of $T$ onto $\mathsf{Lip}_1$ and $\mathbb{R}^d$, respectively. We note that the decoupling here is more delicate than in (Gajjar et al., 2024). In that work, the index set has the product structure $T = \mathsf{Lip}_1 \times B$ with $B$ a ball in $\mathbb{R}^d$, which allows for a clear separation of $f$ and $x$. However, (Gajjar et al., 2024) only obtains a constant-factor approximation.

In contrast, when aiming for a $(1+\epsilon)$-approximation, the only existing argument is that of (Li & Tai, 2025) for the case of a known link function. Adopting this approach leads to index sets $T$ that are not necessarily rectangular; that is, there may not exist sets $X \subseteq \mathsf{Lip}_1$ and $Y \subseteq \mathbb{R}^d$ such that $T = X \times Y$. As a result, the decoupling of $f$ and $x$ cannot be carried out at the level of the index set and will instead be introduced halfway through the chain of inequalities used to bound the associated Rademacher process.

The Dudley's integral corresponding to $\pi_2(T)$ has been handled in (Li & Tai, 2025), whereas controlling the $\pi_1(T)$ component is new to this work and constitutes the main technical challenge that arises from the combination of an unknown link function and a general $\ell_p$-norm. Existing arguments from (Gajjar et al., 2024), which are tailored to the $p = 2$ case, do not generalize easily to other values of $p$.

To address this issue, we develop a simpler and more direct approach for controlling the metric entropy of $\mathsf{Lip}_1$. Rather than explicitly discretizing Lipschitz functions as in (Gajjar et al., 2024), which leads to delicate and technically involved arguments, we abstract the core difficulty into a covering problem of $\mathsf{Lip}_1$ under a family of nonstandard sup-norms. This perspective captures the underlying geometry of the problem while avoiding the technical overhead of explicit discretization and the complicated arguments for controlling set sizes in relation to the discretized functions. The key technical ingredient is the following covering lemma, which may be of independent interest.

**Lemma 1.3** (Informal version of Theorem D.1). *Consider the class* $\mathsf{Lip}_1$ *equipped with a norm defined as the maximum of weighted* $L^\infty$*-norms over possibly different intervals,*

$$\|f\| := \max_{i\in I} \lambda_i \|f\|_{L^\infty([-M_i, M_i])},$$

*where each* $\lambda_i \in (0,1)$ *is a weight. It then holds that*

$$\mathcal{N}(\mathsf{Lip}_1, \|\cdot\|, \epsilon) \leq \mathcal{N}\left(\mathsf{Lip}_1, \|\cdot\|_{L^\infty([-1,1])}, \frac{\epsilon}{CM\ln\kappa}\right),$$

*where* $C > 0$ *is an absolute constant,* $M = \max_{i\in I} \lambda_i M_i$ *and* $\kappa = (\max_{i\in I} M_i)/M$.

We note that the compactness of $\mathsf{Lip}_1$ on bounded domains and standard metric entropy bounds for Lipschitz functions ensure finiteness of covering numbers and, for $p > 1$, the convergence of Dudley's integral. When $p = 1$, the integral does not converge at $0^+$. We therefore control the supremum of the Rademacher process by the sum of a truncated entropy integral and the remaining local oscillation, which results in the same qualitative guarantees as $p > 1$.

Finally, as in (Li & Tai, 2025), a bootstrapping argument is used to achieve a $(1+\epsilon)$-approximation.

**Lower Bound.** Several lower bounds for $\ell_p$-regression and $\ell_p$ subspace embedding rely on the existence of large collections of low-incoherence binary vectors; see, e.g. (Li et al., 2021; Yasuda, 2024). Specifically, for $N = d^{p/2}$, there exist vectors $a_1, \ldots, a_N \in \{\pm 1\}^d$ such that $|\langle a_i, a_j \rangle| \leq C(p)\sqrt{d}$. This construction forms the basis of the corresponding hard instances, and the lower bound in this work is built on the same idea.

By Yao's minimax principle, it suffices to construct a hard distribution over instances with deterministic $f$ and $A$ and a random vector $b$, such that any algorithm that approximately solves the single-index regression problem must make $\Omega(d^{p/2}/\epsilon^p)$ non-adaptive queries to the entries of $b$. Concretely, the random vector $b$ is associated with a uniformly random index $i^* \in [N]$ and a random sign $\sigma^*$ and show that (i) any approximate solution can be used to recover $(i^*, \sigma^*)$ and (ii) identifying $(i^*, \sigma^*)$ with constant probability requires reading $\Omega(d^{p/2}/\epsilon^p)$ entries of $b$.

Our function $f$ is

$$f(t) = \begin{cases} 0, & t \leq T\epsilon\sqrt{d}, \\ t - T\epsilon\sqrt{d}, & T\epsilon\sqrt{d} < t < \epsilon d, \\ \epsilon d - T\epsilon\sqrt{d}, & t \geq \epsilon d, \end{cases} \quad (5)$$

where $T$ is a constant depending only on $p$. Our matrix $A$ is formed by vertically concatenating $2N$ blocks, repeating each vector $a_1, -a_1, \ldots, a_N, -a_N$ for $s = 1/\epsilon^p$ times.

Next we describe the random vector $b$. Initially we set $b = 0$. A uniformly random index $i^* \in [N]$ is selected. We then set all coordinates of $b$ in $(\pm a_{i^*})$-blocks to the value $\epsilon d$. These two blocks are referred to as the planted blocks. Finally, a random sign $\sigma^* \in \{-1, 1\}$ is chosen, determining one of the two planted blocks; a coordinate within the corresponding $(\sigma^* a_{i^*})$-block is then selected and

set to $\epsilon d + d$. This introduces a single *spike* inside the planted blocks. It is immediate that identifying the pair $(i^*, \sigma^*)$ via non-adaptive queries requires reading $\Omega(Ns) = \Omega(d^{p/2}/\epsilon^p)$ coordinates of $b$, which establishes (ii).

We further show that the optimal solution to the single-index regression problem $\min_x \|f(Ax) - b\|_p$ is attained at $\sigma^* \epsilon a_{i^*}$. This follows from the fact that $f(x) \leq \epsilon d - T\epsilon\sqrt{d}$, and so the residuals within the planted blocks can be lower bounded.

Moreover, any good approximate solution $\hat{x}$ must correlate strongly with the optimal solution. Intuitively, if $\hat{x}$ has small correlation with $a_{i^*}$, the residual at the spike is necessarily large; if $\hat{x}$ correlates with some incorrect $a_i$ ($i \neq i^*$), the sum of residuals across the $(\pm a_i)$-blocks will be too large. Consequently, the planted block index $i^*$ can be recovered via $\arg\max_i |\langle a_i, \hat{x} \rangle|$ and the sign $\sigma^*$ via $\mathrm{sgn}\langle a_{i^*}, \hat{x} \rangle$. This establishes (i).

## 2. Preliminaries

**Notation.** We use the shorthand $a \vee b = \max\{a, b\}$ and $a \wedge b = \min\{a, b\}$ for any $a, b \in \mathbb{R}$. We also write $[n] = \{1, \ldots, n\}$ for positive integers $n$.

We write $X \sim \mathcal{D}$ to denote that a random variable $X$ follows the distribution $\mathcal{D}$. For a scalar $c \in \mathbb{R}$, the notation $X \sim c\mathcal{D}$ means that $X$ has the same distribution as $cY$ for $Y \sim \mathcal{D}$. We use $\mathrm{Ber}(\alpha)$ to denote the Bernoulli distribution with parameter $\alpha$, i.e., the distribution of a random variable that equals 1 with probability $\alpha$ and 0 with probability $1 - \alpha$. We use $\mathrm{Bin}(n, \alpha)$ to denote the binomial distribution corresponding to $n$ independent Bernoulli trials with success probability $\alpha$.

For nonnegative functions $f$ and $g$, we write $f \lesssim g$ if $f \leq Cg$ and $f \gtrsim g$ if $f \geq cg$, where $C$ and $c$ are positive constants. When the constant depends on a variable $a$, we write $f \lesssim_a g$ and $f \gtrsim_a g$. We write $f \asymp g$ when both $f \lesssim g$ and $f \gtrsim g$ hold.

**Lewis weights.** Now we recall the definition Lewis weights and some of their basic properties. The forms presented below are from (Li & Tai, 2025).

**Definition 2.1** ($\ell_p$-Lewis weights). Let $A \in \mathbb{R}^{n \times d}$ and $p \geq 1$. For each $i \in [n]$, the $\ell_p$-Lewis weight of $A$ for the $i$-th row is defined to be $w_i$ that satisfies

$$w_i(A) = (a_i^\top (A^\top W^{1 - \frac{2}{p}} A)^\dagger a_i)^{\frac{p}{2}}$$

where $a_i$ is the $i$-th row of $A$ (as a column vector), $W = \mathrm{diag}\{w_1, \ldots, w_n\}$ and $\dagger$ denotes the pseudoinverse.

When the matrix $A$ is clear from context, we simply write $w_i$ instead of $w_i(A)$.

---

**Algorithm 1** $\mathtt{GSM}(k_1, \ldots, k_n, \alpha)$ (Generating Sampling Matrix)

---

**Require:** $n$ integers $k_1, \ldots, k_n \geq 0$; a sampling rate $\alpha < 1$
1: $S \leftarrow$ an $n \times n$ diagonal matrix, initialized to a zero matrix
2: **for** $i = 1, \ldots, n$ **do**
3:     **if** $k_i > 0$ **then**
4:         Generate a binomial variable $N_i \sim \mathrm{Bin}(k_i, \alpha)$
5:         $S_{ii} \leftarrow \left(\frac{N_i}{\alpha k_i}\right)^{\frac{1}{p}}$
6:     **end if**
7: **end for**
8: Return $S$

---

**Algorithm 2** Algorithm for Active Learning

---

**Require:** a matrix $A \in \mathbb{R}^{n \times d}$
        a query access to the entries of the vector $b \in \mathbb{R}^n$
        an error parameter $\epsilon$
        a sampling rate $\alpha < 1$
1: Compute the Lewis weights $w_1, \ldots, w_n$ of $A$
2: **for** $i = 1, \ldots, n$ **do**
3:     $k_i \leftarrow \lceil \frac{n \cdot w_i}{d} \rceil$
4: **end for**
5: $S \leftarrow \mathtt{GSM}(k_1, \ldots, k_n, \alpha)$ from Algorithm 1
6: Solve the minimization problem

$$(\hat{f}, \hat{x}) := \arg\min_{f \in \mathsf{Lip}_1, x \in \mathbb{R}^d} \|Sf(Ax) - Sb\|_p^p + \epsilon\|Ax\|_p^p$$

7: Return $(\hat{f}, \hat{x})$

---

We remark that exact Lewis weights are not needed; constant-factor approximations suffice, increasing the sample complexity by only a constant factor. For simplicity of presentation, however, the algorithms will be stated as if exact Lewis weights were used.

When $p < 4$, constant-factor approximate Lewis weights can be computed in $O(\log \log n)$ rounds of leverage-score estimation (Cohen & Peng, 2015), with each round taking $\tilde{O}(nd + \mathrm{poly}(d))$ time. When $p > 4$, they can be computed in $\tilde{O}(d)$ rounds of leverage-score estimation (Apers et al., 2024). Consequently, the total running times for approximating the Lewis weights are $\tilde{O}(nd + \mathrm{poly}(d))$ and $\tilde{O}(nd^2 + \mathrm{poly}(d))$ in the two regimes, respectively.

## 3. Upper Bound

As previously noted, our approach adopts the framework in (Li & Tai, 2025). The algorithm is presented in Algorithm 2. It is identical to that of (Li & Tai, 2025), with the only difference being that the sketched optimization problem includes minimization over $f$. We briefly describe the

algorithm below.

For the $i$-th row of $A$, let $k_i = \lceil w_i(A)/(d/n) \rceil$ denote its multiplicity. Instead of sampling rows directly with probabilities proportional to their Lewis weights $w_i(A)$, we conceptually "split" the $i$-th row into $k_i$ identical copies. These copies are sampled independently with a fixed global probability $\alpha = \alpha(d, \epsilon, p, n)$. The sampled copies are then re-assembled into a single row and rescaled appropriately, yielding a sketched instance that preserves the original objective value. The algorithm then solves a regularized single-index regression problem of the sketched instance to obtain an approximate solution $(\hat{f}, \hat{x})$.

For the purpose of analysis, it is convenient to work directly with the split instance. Specifically, we replace the $i$-th row of $A$ (and the $i$-th entry of $b$) by $k_i$ identical copies, forming an expanded matrix $A'$ (and vector $b'$). Let $m$ be the total number of rows of $A'$ and define an $m \times m$ diagonal matrix

$$\Lambda = \mathrm{diag}\{\underbrace{k_1^{-1/p}, \ldots, k_1^{-1/p}}_{k_1 \text{ times}}, \ldots, \underbrace{k_n^{-1/p}, \ldots, k_n^{-1/p}}_{k_n \text{ times}}\}.$$

As shown in (Li & Tai, 2025), this construction ensures that

(i) $m \le 2n$,

(ii) (norm preserving) $\|\Lambda(f(A'x) - b')\|_p^p = \|f(Ax) - b\|_p^p$ for all $x \in \mathbb{R}^d$, and

(iii) (uniformly bounded Lewis weights) $w_i(\Lambda A') \le d/n \le 2d/m$ for all $i \in [m]$.

Therefore, the single-index regression problem can be equivalently reformulated using $(A', b')$, without changing either the optimal solution or the objective value. For notational simplicity, in the remainder of the paper we continue to write $A$ and $b$, with the understanding that they have been preprocessed to satisfy the properties above.

Once uniformly bounded Lewis weights (of $\Lambda A$) are in place, the algorithm simply samples the coordinates of $b$ independently with probability $\alpha$ and rescales by $\alpha^{-1/p}$. This corresponds to multiplying $f(Ax)$ and $b$ by a diagonal matrix $S$ in which $S_{ii}$ are i.i.d. $\alpha^{-1/p} \mathrm{Ber}(\alpha)$ variables.

Analogous to (Li & Tai, 2025), our main theorem in the analysis is the following.

**Theorem 3.1.** *Let $p > 1$ be a constant, $A \in \mathbb{R}^{n \times d}$, $\epsilon \in (0,1)$ be sufficiently small, and $\Lambda \in \mathbb{R}^{n \times n}$ be a diagonal matrix with entries $\Lambda_{ii} > 0$ such that $w_i(\Lambda A) \lesssim d/n$ for all $i \in [n]$. Let $v \in \mathbb{R}^n$ be a fixed vector, $(\bar{f}, \bar{x}) \in \mathsf{Lip}_1 \times \mathbb{R}^d$ be a fixed reference point, and $V := \|\Lambda(\bar{f}(A\bar{x}) - v)\|_p^p$.*

*Let $R \ge \|\Lambda A\bar{x}\|_p^p$ and $F \ge V$ be fixed upper bounds. Suppose the set $T$ satisfies $\{(\bar{f}, \bar{x})\} \subseteq T \subseteq \{(f, x) \in \mathsf{Lip}_1 \times \mathbb{R}^d : \|\Lambda Ax\|_p^p \le R \text{ and } \|\Lambda(f(Ax) - \bar{f}(A\bar{x}))\|_p^p \le F\}$.*

*Let $\alpha \in [0,1]$ and $S$ be an $n \times n$ random diagonal matrix with i.i.d. $\alpha^{-1/p} \mathrm{Ber}(\alpha)$ entries. Conditioned on the event that*

$$\|S\Lambda(\bar{f}(A\bar{x}) - v)\|_p^p \lesssim V$$

*and*

$$\sup_{(f,x) \in T} \|S\Lambda(f(Ax) - \bar{f}(A\bar{x}))\|_p^p \le F,$$

*it holds with probability at least $1 - \delta$ that*

$$\sup_{(f,x) \in T} \left| \left( \|S\Lambda(f(Ax) - v)\|_p^p - \|\Lambda(f(Ax) - v)\|_p^p \right) \right.$$
$$\left. - \left( \|S\Lambda(\bar{f}(A\bar{x}) - v)\|_p^p - \|\Lambda(\bar{f}(A\bar{x}) - v)\|_p^p \right) \right|$$
$$\le C \left( \epsilon V + \frac{d^{1 \vee \frac{p}{2}}}{\alpha n} R + \Gamma \ln \frac{n}{\epsilon d} \left( \ln^{\frac{5}{4}} d + \sqrt{\ln \frac{1}{\delta}} \right) \right),$$

*where $C$ depends only on $p$, and $\Gamma$ is defined as*

$$\Gamma := \begin{cases} d^{\frac{1}{2}}(\alpha n)^{-\frac{1}{2}} F^{\frac{1}{2}} R^{\frac{1}{2}}, & 1 \le p \le 2 \\ d^{\frac{1}{2}}(\alpha n)^{-\frac{1}{p}} F^{1 - \frac{1}{p}} R^{\frac{1}{p}}, & p > 2. \end{cases} \quad (6)$$

The proof of Theorem 3.1 is lengthy and technically involved. As outlined in Section 1.2, our analysis builds upon the approach in (Li & Tai, 2025), but introduces a key innovation: we bound Dudley's integral over $\mathsf{Lip}_1$ by deriving a new bound on the covering number of $\mathsf{Lip}_1$ under the supremum of several $L^\infty$-norms over different intervals. The full proof of Theorem 3.1 appears in Appendices B to E, culminating in the proof of the theorem in Appendix E. When $p = 1$, an analogous version of Theorem 3.1 also holds, with $\log^{\frac{5}{4}} d$ replaced with $\log^{\frac{5}{4}} d + \log n$. The proof is provided in Appendix G.

Given Theorem 3.1 (together with its $p = 1$ analogue), the mixed-error guarantee for the solution $(\hat{f}, \hat{x})$ of the sketched single-index problem follows via a near-identical bootstrapping argument used in (Li & Tai, 2025). This leads to the following theorem, proved in Appendix F.

**Theorem 3.2.** *Let $p \ge 1$ be a constant, $A \in \mathbb{R}^{n \times d}$, $b \in \mathbb{R}^n$, $\epsilon \in (0,1)$ be sufficiently small and $\Lambda$ be an $n \times n$ diagonal matrix satisfying $\Lambda_{ii} > 0$ and $w_i(\Lambda A) \lesssim d/n$ for all $i$. There exists a randomized algorithm which, with probability at least $0.9$, makes $O(d^{1 \vee \frac{p}{2}} / \epsilon^{p \vee 2} \operatorname{poly} \log n)$ queries to the entries of $b$ and returns $(\hat{f}, \hat{x}) \in \mathsf{Lip}_1 \times \mathbb{R}^d$ satisfying the mixed error guarantee (3). The hidden constants in the bounds on the number of queries depends on $p$ only.*

We leave open the question of whether the $\log n$ factors can be removed. In (Li & Tai, 2025), those factors come from Dudley's integral and the authors circumvent them by first using a net argument that leads to a sub-optimal complexity of $\operatorname{poly}(d/\epsilon)$ queries, which effectively reduces

$n$ to $\mathrm{poly}(d/\epsilon)$. In our case, however, the covering number for Lipschitz functions intrinsically introduces a $\log n$ factor (see Lemma 1.3), which a net argument in place of chaining cannot remove.

# 4. Non-adaptive Lower Bound

As previewed in Section 1.2, by Yao's minimax principle, it suffices to construct a deterministic matrix $A$, a deterministic function $f$, and a random $b$ that encodes an index $i^* \in [N]$ and a sign $\sigma^* \in \{-1, 1\}$ such that a good approximation solution to the single-index regression allows the recovery of $i^*$ and $\sigma^*$. Below we describe our hard instance.

**Setup.** Let $p > 2$, $\epsilon \in (0, 1)$, $N = d^{p/2}$ and $s = \epsilon^{-p}$. Following (Parampalli et al., 2013), there exist $a_1, \dots, a_N \in \{\pm 1\}^d$ which satisfy $|\langle a_i, a_j \rangle| \leq C\sqrt{d}$ for all $i \neq j$, where $C$ is a constant depending only on $p$. The function $f$ is defined as in (5) with $T = C$. It is clear that $f$ is increasing and 1-Lipschitz. We also assume that $d \geq 4T^2$.

The matrix $A \in \mathbb{R}^{(2Ns) \times d}$ is constructed by tiling $s$ repeated copies of each $a_i$ and $-a_i$ as rows:

$$A_{j,*} = \begin{cases} a_i, & j = (2i-2)s+1, \dots, (2i-1)s, \\ -a_i, & j = (2i-1)s+1, \dots, 2is. \end{cases}$$

We refer to rows $(2i-2)s+1, \dots, (2i-1)s$ as the $(+a_i)$-block and rows $(2i-1)s+1, \dots, 2is$ as the $(-a_i)$-block.

**Distribution of $b$.** Consider the following distribution $\mathcal{D}_\in$ of $b$. Let $i^* \sim \mathrm{Unif}([N])$, $\sigma^* \sim \mathrm{Unif}(\{\pm 1\})$ and $k^* \sim \mathrm{Unif}([s])$. Let $j^* = (2i^* - \frac{\sigma^*+1}{2})s + k^*$ denote the index of a unique spike. The vector $b \in \mathbb{R}^{2Ns}$ is defined by

$$b_j = \begin{cases} \epsilon d + d, & j = j^*, \\ \epsilon d, & (2i^*-2)s+1 \leq j \leq 2i^*s \text{ and } j \neq j^*, \\ 0, & \text{otherwise.} \end{cases}$$

Intuitively, all coordinates in the $(\pm a_{i^*})$-blocks (the *planted blocks*) receive a background value $\epsilon d$, except for a single *spike coordinate*, whose value is $\epsilon d + d$. The spike may fall in the positive block or the negative block.

It is not difficult to see that recovering $i^*$ and $\sigma^*$ requires $\Omega(Ns) = \Omega(d^{p/2}/\epsilon^p)$ queries to $b$ in the non-adaptive setting. This is formalized in the lemma below. All proofs in this section are deferred to Appendix H.

**Lemma 4.1.** *Let $\mathcal{A}$ be a randomized algorithm that takes* nonadaptive *samples of $b$: it fixes a set $S \subseteq [2Ns]$ of indices with $|S| = q$, reads $\{b_j : j \in S\}$ and outputs $(\hat{i}, \hat{\sigma})$. If $\Pr\{(\hat{i}, \hat{\sigma}) = (i^*, \sigma^*)\} \geq 2/3$, then $q \geq (2Ns)/3$.*

The main result is to show that $i^*$ and $\sigma^*$ can be recovered from a solution $\hat{x}$ that satisfies the error guarantee (3).

**Theorem 4.2.** *Let $K = 4^p(C^p+1)$. Suppose that $\hat{x}$ satisfies*

$$\|f(A\hat{x}) - b\|_p^p \leq \left(1 + \frac{\epsilon}{K}\right)\|f(Ax^*) - b\|_p^p + \frac{\epsilon}{K}\|Ax^*\|_p^p,$$

*then it holds that $i^* = \arg\max_i |\langle a_i, \hat{x}\rangle|$ and $\mathrm{sgn}\langle a_{i^*}, \hat{x}\rangle = \sigma^*$.*

The proof proceeds as follow. First, we verify that $x^* = \sigma^*\epsilon a_i$ is indeed the minimizer of the objective function $\Phi(x) = \|f(Ax) - b\|_p$ by explicitly calculating $\Phi(\sigma^*\epsilon a_i)$ and $\Phi(-\sigma^*\epsilon a_i)$. Second, we calculate $\|Ax^*\|_p^p$ and use the error guarantee (3) to show that $|\langle a_i, \hat{x}\rangle|$ must be large for $i = i^*$ and small for all $i \neq i^*$. Finally, we evaluate the gap between $\Phi(x^*)$ and $\Phi(-x^*)$ to show that the sign can be recovered.

The following non-adaptive sampling lower bound is an immediate corollary of Lemma 4.1 and Theorem 4.2.

**Theorem 4.3** (Non-adaptive query lower bound). *Suppose that $p \geq 2$ is a constant, $\epsilon > 0$ is sufficiently small, $d$ is at least some constant depending only on $p$ and $n \gtrsim_p d^{p/2}/\epsilon^p$. There exist a deterministic function $f \in \mathsf{Lip}_1$, a deterministic matrix $A \in \mathbb{R}^{n \times d}$ and a distribution over $b \in \mathbb{R}^n$ such that the following holds: every deterministic algorithm that outputs $\hat{x} \in \mathbb{R}^d$ which with probability at least $4/5$ over the randomness of $b$ satisfies (3) must make $\Omega(d^{p/2}/\epsilon^p)$ non-adaptive queries to the entries of $b$.*

We remark that the hard instance constructed above can be solved with much fewer queries in the adaptive setting. In particular, one can first read a single coordinate from each block of $b$ to identify the planted index $i^*$. Subsequently, by querying all $2/\epsilon^p$ entries in the $(+a_{i^*})$- and $(-a_{i^*})$-blocks, one can recover the sign $\sigma^*$. This procedure uses only $\Theta(d^{p/2} + 1/\epsilon^p)$ adaptive queries. We conjecture the lower bound $\Omega(d^{p/2}/\epsilon^p)$ continues to hold in the adaptive setting; proving such a bound remains an open question.

# 5. Experiments

We conduct the experiment on both synthetic data and real-world data.

**Synthetic Data** Define ReLU link functions $f_{\alpha,\tau}(x) = \alpha\max\{0, x - \tau\}$, with $\alpha \in [0, 1]$. The true parameters are $\alpha^* = 0.8$ and $\tau^* = 0.5$. For the data generation, we set $n = 2000$, $d = 20$, and fix $x^*$ to be a unit vector. The rows $a_i$ of the matrix $A \in \mathbb{R}^{n \times d}$ are independent and identically distributed from the following mixture distribution

$$a_i \sim \begin{cases} N(8x^*, 100\Sigma) & \text{with probability } 0.05, \\ N(0, \Sigma) & \text{with probability } 0.95, \end{cases}$$

where the covariance matrix $\Sigma \in \mathbb{R}^{d \times d}$ is diagonal,

i.e.,

$$\Sigma = \text{diag}(\sigma_1^2, \ldots, \sigma_d^2)$$

with

$$\sigma_j^2 = 1 - \frac{j-1}{d-1}(0.9).$$

The vector $b$ is defined as $b_i = f_{\alpha^*, \tau^*}(a_i^\top x^*) + \epsilon_i$, where $\epsilon_i$ are i.i.d. $N(0, 0.05^2)$ variables.

**Real-world Data** We use the Communities and Crime dataset from the UCI Machine Learning Repository. The Communities and Crime dataset[1] (referred to as *Communities*) and has dimension $1994 \times 127$. We remove one categorical variable and use the remaining numerical features in all experiments. This dataset has also been used as a benchmark in prior work on single-index models; see, e.g., (Kakade et al., 2011).

We shall empirically verify that Algorithm 2 generates good approximations to the optimizers, with fast runtime, for active regression in single-index models with an unknown link function, for both $p = 1$ and $p = 3$.

For the synthetic dataset, we restrict the function $f$ to the ReLU family used in the data-generation process. For the Communities dataset, we optimize over the broader class of increasing 1-Lipschitz functions satisfying $f(0) = 0$. Below, we describe the optimization procedure for the real-world dataset, which is based on the Isotron framework; the synthetic case is simpler, obtained by replacing the Isotron step with direct optimization over the ReLU parameters.

**Solving the Full-scale Problem** Recall that the inputs are a matrix $A \in \mathbb{R}^{n \times d}$ and a label vector $b \in \mathbb{R}^n$. Fixing $p$, we solve the full optimization problem $\min_{f,x} \|f(Ax) - b\|_p^p$, where $f$ is increasing, 1-Lipschitz and satisfies $f(0) = 0$, obtaining $(f^*, x^*)$ together with its running time. Using this solution, we compute $\mathsf{OPT} = \|f^*(Ax^*) - b\|_p^p$ and $\|Ax^*\|_p^p$.

The algorithm we use to solve for the optimization problem is a variant of the Isotron algorithm. To the best of our knowledge, the original isotron framework for learning single-index models with an unknown link function was introduced in (Kalai & Sastry, 2009), with subsequent refinements and extensions in (Kakade et al., 2011), (Gollakota et al., 2023) and (Zarifis et al., 2024). More recently, Hu et al. (2025) study single-index models in the context of omniprediction. All of these works focus on $p = 2$. Our focus here is to use the Isotron framework as an efficient numerical method for general $p$, without providing new theoretical guarantees.

---

[1] https://archive.ics.uci.edu/dataset/183/communities+and+crime

In our Isotron variant, we solve a regularized regression problem

$$\min_{f,x} \|\Lambda^{1/p}(f(Ax) - b)\|_p^p + \epsilon_{\text{reg}} \|\Lambda^{1/p} Ax\|_p^p$$

as required in Algorithm 2, where $\Lambda$ is a diagonal matrix of positive rescaling factors. One can set $\epsilon_{\text{reg}} = 0$ and $\Lambda = I$ when solving the full-scale problem. Each iteration alternates between two steps.

1. Given the iterate $x^{(t)}$ in the $t$-th iteration, we form $z = Ax^{(t)}$ and solve for the optimal link function values $u \in \mathbb{R}^n$ on the $z$ points by minimizing

$$\min_{u \in \mathbb{R}^n} \sum_{i=1}^n \lambda_i |u_i - b_i|^p$$
$$\text{s.t.} \quad 0 \leq u_{i+1} - u_i \leq z_{i+1} - z_i,$$

   after sorting the indices of $z$. This problem is convex and we solve it using projected gradient descent, where the projection is implemented via Pool Adjacent Violators Algorithm (PAVA) for monotonicity and Dykstra's algorithm to enforce the intersection with the Lipschitz constraint. The constraint $f(0) = 0$ is enforced by adding a point $(z_0, u_0) = (0, 0)$. This step yields a piecewise linear, increasing function $f^{(t)} \in \text{Lip}_1$.

2. Treating the fitted link function $f^{(t)}$ as fixed, we update $x$ by a gradient step on the regularized objective

$$\|f^{(t)}(Ax) - b\|_p^p + \epsilon_{\text{reg}} \|Ax\|_p^p.$$

   Specifically, letting $r = f^{(t)}(Ax^{(t)}) - b$ and $z = Ax^{(t)}$, we perform the update

$$x^{(t+1)} = x^{(t)} - \eta A^\top \Lambda \left( \nabla_r \|r\|_p^p + \epsilon_{\text{reg}} \nabla_z \|z\|_p^p \right),$$

   where $\nabla_r \|r\|_p^p$ is a vector defined as

$$(\nabla_r \|r\|_p^p)_i = \begin{cases} p \, \text{sgn}(r_i) |r_i|^{p-1}, & p > 1 \\ \text{sgn}(r_i), & p = 1. \end{cases}$$

We run the algorithm for 50 iterations and return the iterate $(f, x)$ that achieves the smallest objective value as $(f^*, x^*)$.

**Solving the Sketched Problem** Following our main algorithm (Algorithm 2), for the sketched problem, we compute an approximate solution $(\hat{f}, \hat{x})$ by solving

$$\min_{f,x} \|\Lambda^{1/p}(f(A'x) - b')\|_p^p + \epsilon_{\text{reg}} \|\Lambda^{1/p} A'x\|_p^p,$$

where $A'$ and $b'$ consist of a randomly sampled subset of rows of on a randomly sampled subset of rows of $A$ and $b$, $\Lambda$ is a diagonal matrix of rescaling factors (based on sampling weights), and $\epsilon_{\text{reg}}$ is a regularization parameter.

*Table 1.* Runtimes (in seconds) under Lewis weight sampling on the Communities dataset, for $p = 1$ (top) and $p = 3$ (bottom). Speedup is computed as the ratio between the median runtime on the full dataset and the median runtime of the corresponding sketch.

| Sample fraction | 25th pct. | Median | 75th pct. | Speedup |
|---|---|---|---|---|
| 0.10 | 4.956 | 5.224 | 5.346 | 126.3× |
| 0.15 | 7.659 | 7.862 | 7.944 | 83.9× |
| 0.20 | 10.540 | 10.676 | 10.734 | 61.8× |
| 0.25 | 13.228 | 13.661 | 13.728 | 48.3× |
| 0.30 | 16.009 | 16.294 | 16.411 | 40.5× |
| Full dataset (100% rows) | | | | 659.961 |

| Sample fraction | 25th pct. | Median | 75th pct. | Speedup |
|---|---|---|---|---|
| 0.10 | 5.068 | 5.177 | 5.299 | 126.0× |
| 0.15 | 7.430 | 7.710 | 7.845 | 84.6× |
| 0.20 | 9.302 | 10.054 | 10.253 | 64.9× |
| 0.25 | 12.049 | 12.489 | 12.738 | 52.2× |
| 0.30 | 14.011 | 14.768 | 15.285 | 44.2× |
| Full dataset (100% rows) | | | | 652.508 |

Our algorithm uses Lewis weight sampling to construct $A'$ and $b'$; we compare this approach against the baseline of uniform sampling. The sketched problem is solved using the same Isotron-based algorithm described above. The goal of these experiments is to compare the error and runtime (achievable speedup) under different sampling strategies and sketch sizes.

For each sampling scheme and sampling size (specified as a fraction of full dataset), we report the smallest $\epsilon$ such that

$$\|\hat{f}(A\hat{x}) - b\|_p^p \leq (1 + \epsilon)\mathsf{OPT} + \epsilon\|Ax^*\|_p^p.$$

We report the median and inter-quartiles of both runtime and $\epsilon$ over 10 independent trials.

In our experiments, we consider $p \in \{1, 3\}$ and vary the sampling fraction from $0.1$ to $0.3$. We set the regularization parameter to $\epsilon_{\mathrm{reg}} = 0.25$ for $p = 1$ and $\epsilon_{\mathrm{reg}} = 0.1$ for $p = 3$. These values are chosen so that $\mathsf{OPT}$ and $\|Ax^*\|_p^p$ are of comparable magnitude, ensuring that the resulting approximation errors $\epsilon$ are meaningful. We emphasize that the purpose of these experiments is to empirically validate the behaviour of the proposed algorithm and its approximation guarantees, rather than to propose a fully parameter-free method.

**Results** For the synthetic dataset, the approximation parameter $\epsilon$ is reported in Figure 1. Lewis weight sampling consistently achieves smaller values of $\epsilon$ than uniform sampling across most sampling fractions. More importantly, it also exhibits noticeably smaller interquartile ranges, indicating more stable approximation quality at comparable sketch sizes. The advantage of Lewis weight sampling is particularly pronounced when $p = 3$.

For the Communities dataset, the corresponding results are

*Figure 1.* Comparison of $\epsilon$ from the sketched problem under different sampling schemes on the synthetic dataset, for $p = 1$ and $p = 3$.

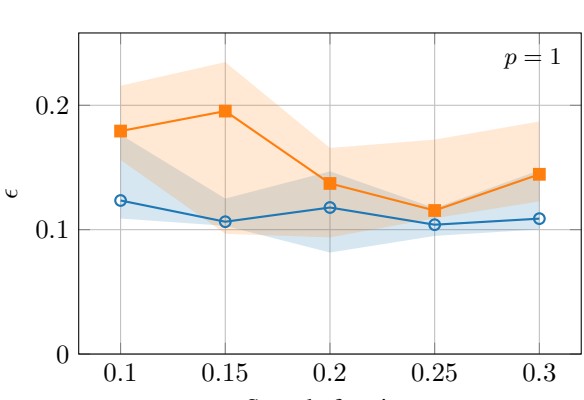

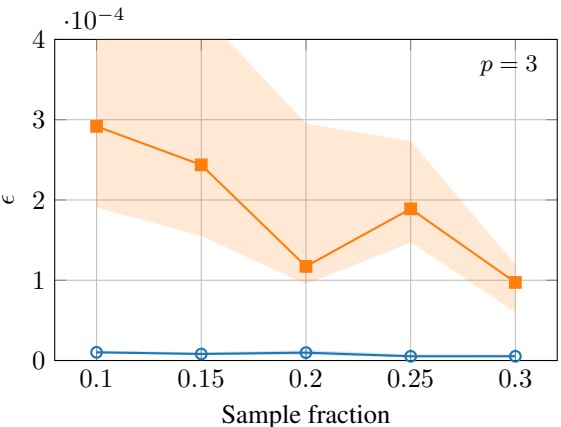

shown in Figure 2. As expected, for Lewis-weight sampling, $\epsilon$ decreases steadily as the sample size increases. Moreover, Lewis-weight sampling again consistently outperforms uniform sampling across most sampling fractions, both in terms of the median value of $\epsilon$ and the interquartile range.

The results of running times for Lewis weight sampling on the Communities dataset are reported in Table 1. Solving the sketched problem is substantially faster than solving the full optimization problem. Since the running times obtained using uniform sampling are comparable across all settings, we omit them from the table to avoid redundancy.

## Acknowledgements

We thank the anonymous reviewers for their suggestions regarding the presentation of the paper and the experiments on synthetic datasets.

C.W. In is supported by an NTU Research Scholarship. Y. Li is supported in part by the Singapore Ministry of Education

*Figure 2.* Comparison of $\epsilon$ from the sketched problem under different sampling schemes on the Communities dataset, for $p = 1$ and $p = 3$.

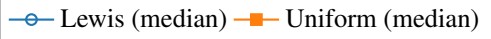

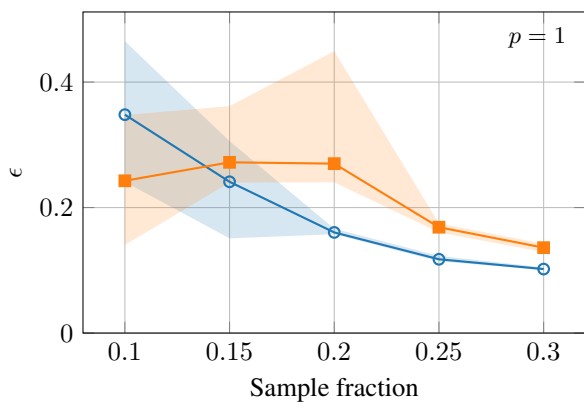

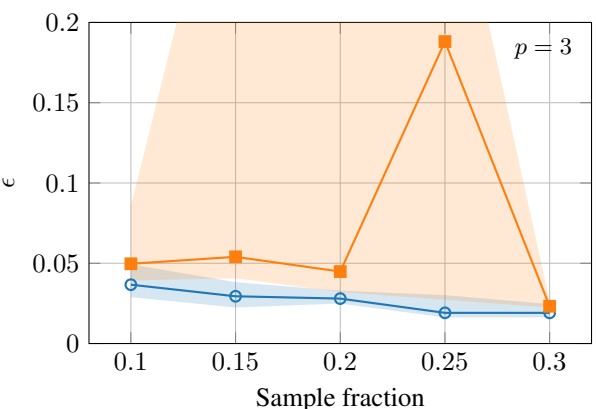

AcRF Tier 1 grant RG21/25. X. Wu was supported by the Singapore Ministry of Education AcRF Tier 1 grant RG21/25 while he was at Nanyang Technological University, where part of this work was done.

## Impact Statement

This paper presents work whose goal is to advance the field of Machine Learning. There are potentially societal consequences of our work, none which we feel must be specifically highlighted here.

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

# A. Additional Preliminaries

**Covering Number and Dudley's Integral.** Suppose that $(X, d)$ is a pseudometric space. The $\epsilon$-covering number of $X$, denoted by $\mathcal{N}(X, d, \epsilon)$, is the minimum $m$ such that there exist $m$ points $x_1, \ldots, x_m \in X$ such that for every $x \in X$, there exist $j \in [m]$ such that $d(x, x_j) < \epsilon$.

Let $(X_t)_{t \in T}$ be a zero-mean subgaussian process with metric $d$ on a set $T$. It is a classical result (see, e.g. (Vershynin, 2018)) that

$$\left[ \mathbb{E} \left( \sup_{s,t \in T} |X_t - X_s| \right)^\ell \right]^{\frac{1}{\ell}} \lesssim \int_0^\infty \sqrt{\ln \mathcal{N}(T, d, \epsilon)} d\epsilon + \sqrt{\ell} \operatorname{diam}(T, d). \tag{7}$$

When the covering number $\sqrt{\ln \mathcal{N}(T, d, \epsilon)}$ increases too rapidly as $\epsilon \to 0^+$, we have

$$\left[ \mathbb{E} \left( \sup_{s,t \in T} |X_t - X_s| \right)^\ell \right]^{\frac{1}{\ell}} \lesssim \left[ \mathbb{E} \left( \sup_{\substack{s,t \in T \\ d(s,t) \leq \epsilon_0}} |X_t - X_s| \right)^\ell \right]^{\frac{1}{\ell}} + \int_{\epsilon_0/2}^\infty \sqrt{\ln \mathcal{N}(T, d, \epsilon)} d\epsilon + \sqrt{\ell} \operatorname{diam}(T, d). \tag{8}$$

We shall use the former form for $p > 1$ and the latter one for $p = 1$.

**Rademacher Processes.** Let $\{\xi_i\}_{i=1}^n$ be independent Rademacher random variables, and let $T \subset \mathbb{R}^n$. The associated Rademacher process $\{X_t\}_{t \in T}$ on $T$ is defined as

$$X_t := \sum_{i=1}^n \xi_i t_i = \langle \xi, t \rangle.$$

The canonical (pseudo-)metric of this process is

$$d(s, t) := (\mathbb{E}(X_s - X_t)^2)^{1/2} = \|s - t\|_2, \qquad s, t \in T.$$

Moreover, the process $\{X_t\}_{t \in T}$ is a subgaussian process with respect to the metric $d$.

**Lewis weights.** First we present a lemma on the basic properties of Lewis weight (Li & Tai, 2025).

**Lemma A.1.** *Suppose that $A \in \mathbb{R}^{n \times d}$ has full column rank and Lewis weights $w_1, \ldots, w_n$. Let $W = \operatorname{diag}\{w_1, \ldots, w_n\}$. The following properties hold.*

*(1)* $\sum_i w_i = d$;

*(2) There exists a matrix $U \in \mathbb{R}^{n \times d}$ such that*

    *(a) the column space of $U$ is the same as that of $A$;*
    *(b) $w_i = \|U_{i,*}\|_2^p$, where $U_{i,*}$ denotes the $i$-th row of $U$;*
    *(c) $W^{\frac{1}{2} - \frac{1}{p}} U$ has orthonormal columns;*

*(3) It holds for all vectors $u$ in the column space of $A$ that $\|W^{\frac{1}{2} - \frac{1}{p}} u\|_2 \leq d^{\frac{1}{2} - \frac{1}{2 \vee p}} \|u\|_p$.*

*(4) It holds for all vectors $u$ in the column space of $A$ that $|u_i| \leq d^{\frac{1}{2} - \frac{1}{2 \vee p}} w_i^{\frac{1}{p}} \|u\|_p$.*

We present some properties regarding $\Lambda_{ii}(f_1(Ax_1) - f_2(Ax_2))_i$.

**Lemma A.2.** *For any $R \geq \|\Lambda A \bar{x}\|_p^p$, let $T = \{x \in \mathbb{R}^d : \|\Lambda A x\|_p^p \leq R\}$. Also, let $w_1, \ldots, w_n$ be Lewis weights of $\Lambda A$. It holds for all $x, \bar{x} \in T$, $f, \bar{f} \in \mathsf{Lip}_1$ and $i \in [n]$ that*

$$|\Lambda_{ii}(f(Ax) - f(A\bar{x}))_i| \leq 2 d^{\frac{1}{2} - \frac{1}{2 \vee p}} w_i^{\frac{1}{p}} R^{\frac{1}{p}},$$

$$|\Lambda_{ii}(f(Ax) - \bar{f}(Ax))_i| \leq 2 d^{\frac{1}{2} - \frac{1}{2 \vee p}} w_i^{\frac{1}{p}} R^{\frac{1}{p}}.$$

*Consequently, by the triangle inequality,*

$$|\Lambda_{ii}(f(Ax) - \bar{f}(A\bar{x}))_i| \leq 4 d^{\frac{1}{2} - \frac{1}{2 \vee p}} w_i^{\frac{1}{p}} R^{\frac{1}{p}}.$$

*Proof.* By the Lipschitz condition and by Lemma A.1(4),

$$|\Lambda_{ii}(f(Ax) - f(A\bar{x}))_i| \leq |\Lambda_{ii}(Ax - A\bar{x})_i| \leq d^{\frac{1}{2} - \frac{1}{2\vee p}} w_i^{\frac{1}{p}} \|\Lambda Ax - \Lambda A\bar{x}\|_p.$$

Since both $x, \bar{x} \in T$, we have

$$\|\Lambda Ax - \Lambda A\bar{x}\|_p \leq \|\Lambda Ax\|_p + \|\Lambda A\bar{x}\|_p \leq 2R^{\frac{1}{p}}.$$

This proves the first inequality.

For the second inequality, since $f, \bar{f} \in \mathsf{Lip}_1$, we know that $f - \bar{f} \in \mathsf{Lip}_2$ and, by Lemma A.1(4) again,

$$|\Lambda_{ii}(f(Ax) - \bar{f}(Ax))_i| \leq 2|\Lambda_{ii}(Ax)_i| \leq 2d^{\frac{1}{2} - \frac{1}{2\vee p}} w_i^{\frac{1}{p}} \|\Lambda Ax\|_p \leq 2d^{\frac{1}{2} - \frac{1}{2\vee p}} w_i^{\frac{1}{p}} R^{\frac{1}{p}}.$$

This proves the second inequality. $\qquad\square$

**Moment bounds for the binomial variables**  We shall need an upper bound on higher moments of binomial variables.

**Lemma A.3** ((Ahle, 2022)). *Let $X \sim \mathrm{Bin}(n, p)$ be a binomial random variable with mean $\mu = np$. Then for every integer $\ell \geq 1$,*

$$\mathbb{E}\, X^\ell \leq \left(\mu + \frac{\ell}{2}\right)^\ell.$$

## B. Formulation of the Error Bound and Analysis for Upper Bound

Suppose that $T$ is a subset of $\mathsf{Lip}_1 \times \mathbb{R}^d$. Let $\pi_1$ denote the projection on $\mathsf{Lip}_1$ and $\pi_2$ the projection on $\mathbb{R}^d$.

Recall that we need to bound the following error:

$$\Psi := \sup_{(f,x)\in T} \left| (\|S\Lambda(f(Ax) - v)\|_p^p - \|\Lambda(f(Ax) - v)\|_p^p) - (\|S\Lambda(\bar{f}(A\bar{x}) - v)\|_p^p - \|\Lambda(\bar{f}(A\bar{x}) - v)\|_p^p) \right|, \qquad (9)$$

which can be rewritten as

$$\Psi = \sup_{(f,x)\in T} \left| \sum_{i=1}^n (S_{ii}^p - 1)(|(f(Ax) - v)_i|^p - |(\bar{f}(A\bar{x}) - v)_i|^p) \right|. \qquad (10)$$

Similar to the approach in (Li & Tai, 2025), we partition the index set $[n]$ into three disjoint sets $J$, $G \setminus J$ and $G^c$, where

$$G := \left\{ i \in [n] : |\Lambda_{ii}(\bar{f}(A\bar{x}) - v)_i| \leq \frac{d^{\frac{1}{2} - \frac{1}{2\vee p}} w_i^{\frac{1}{p}} R^{\frac{1}{p}}}{\epsilon} \right\}, \quad J := \left\{ i \in G : w_i > \frac{\epsilon^p d}{n^2} \right\}.$$

Correspondingly, we split the summation over $[n]$ in (10) into three sums, and consequently,

$$\Psi \leq \Psi_1 + \Psi_2 + \Psi_3,$$

where

$$\Psi_1 = \sup_{(f,x)\in T} \left| \sum_{i \notin G} (S_{ii}^p - 1)\Lambda_{ii}^p (|(f(Ax) - v)_i|^p - |(\bar{f}(A\bar{x}) - v)_i|^p) \right|$$

$$\Psi_2 = \sup_{(f,x)\in T} \left| \sum_{i \in G\setminus J} (S_{ii}^p - 1)\Lambda_{ii}^p (|(f(Ax) - v)_i|^p - |(\bar{f}(A\bar{x}) - v)_i|^p) \right|$$

$$\Psi_3 = \sup_{(f,x)\in T} \left| \sum_{i \in J} (S_{ii}^p - 1)\Lambda_{ii}^p (|(f(Ax) - v)_i|^p - |(\bar{f}(A\bar{x}) - v)_i|^p) \right|$$

Bounding $\Psi_1$ and $\Psi_2$ is relatively straightforward; the arguments follow those in (Li & Tai, 2025), though they are slightly more involved due to the varying functions $f$.

**Lemma B.1** (Bounding $\Psi_1$). *For any $R \geq \|\Lambda A\bar{x}\|_p^p$ and $\epsilon > 0$, let $T$ be a set such that $\{\bar{x}\} \subseteq T \subseteq \{x \in \mathbb{R}^d : \|\Lambda Ax\|_p^p \leq R\}$. Suppose that $S$ is an $n$-by-$n$ diagonal matrix with non-negative diagonal entries and*

$$\|S\Lambda(\bar{f}(A\bar{x}) - v)\|_p^p \lesssim V,$$

*where $V = \|\Lambda \bar{f}(A\bar{x}) - v\|_p^p$. Then we have*

$$\Psi_1 \lesssim \epsilon V.$$

*Proof.* By the triangle inequality,

$$\left| \sum_{i \notin G} (S_{ii}^p - 1)\Lambda_{ii}^p (|(f(Ax) - v)_i|^p - |(\bar{f}(A\bar{x}) - v)_i|^p) \right|$$

$$\leq \sum_{i \notin G} (S_{ii}^p + 1)\Lambda_{ii}^p \left| |(f(Ax) - v)_i|^p - |(\bar{f}(A\bar{x}) - v)_i|^p \right|.$$

Using the inequality $||a|^p - |b|^p| \leq p|a - b|(|a|^{p-1} + |b|^{p-1})$, we have

$$\Lambda_{ii}^p \left| |(f(Ax) - v)_i|^p - |(\bar{f}(A\bar{x}) - v)_i|^p \right|$$

$$\leq p |\Lambda_{ii}(f(Ax) - \bar{f}(A\bar{x}))_i| (|\Lambda_{ii}(f(Ax) - v)_i|^{p-1} + |\Lambda_{ii}(\bar{f}(A\bar{x}) - v)_i|^{p-1}).$$

Hence, for $i \notin G$, by Lemma A.2 and the definition of $G$, it holds that

$$|\Lambda_{ii}(f(Ax) - \bar{f}(A\bar{x}))_i| \leq 4d^{\frac{1}{2} - \frac{1}{2\vee p}} w_i^{\frac{1}{p}} R^{\frac{1}{p}} \leq 4\epsilon |\Lambda_{ii}(\bar{f}(A\bar{x}) - v)_i|$$

and so

$$|\Lambda_{ii}(f(Ax) - v)_i| \leq |\Lambda_{ii}(f(Ax) - \bar{f}(A\bar{x}))_i| + |\Lambda_{ii}(\bar{f}(A\bar{x}) - v)_i| \leq (1 + 4\epsilon)|\Lambda_{ii}(\bar{f}(A\bar{x}) - v)_i|.$$

It follows that

$$\Lambda_{ii}^p |(f(Ax) - v)_i|^p - |(\bar{f}(A\bar{x}) - v)_i|^p| \lesssim \epsilon |\Lambda_{ii}(\bar{f}(A\bar{x}) - v)_i|^p.$$

Therefore,

$$\left| \sum_{i \notin G} (S_{ii}^p - 1)\Lambda_{ii}^p (|(f(Ax) - v)_i|^p - |(\bar{f}(A\bar{x}) - v)_i|^p) \right| \lesssim \epsilon \sum_{i \notin G} (S_{ii}^p + 1)|\Lambda_{ii}(\bar{f}(A\bar{x}) - v)_i|^p$$

$$\leq \epsilon \sum_{i=1}^n (S_{ii}^p + 1)|\Lambda_{ii}(\bar{f}(A\bar{x}) - v)_i|^p \lesssim \epsilon V. \qquad \square$$

**Lemma B.2** (Bounding $\Psi_2$). *For any $R \geq \|\Lambda A\bar{x}\|_p^p$ and $\epsilon > 0$, let $T$ be a set that $\{\bar{x}\} \subseteq T \subseteq \{x \in \mathbb{R}^d : \|\Lambda Ax\|_p^p \leq R\}$. Suppose $S \in \mathbb{R}^{n \times n}$ is a diagonal matrix whose entries satisfy $0 \leq S_{ii}^p \leq \frac{1}{\alpha}$ for any $\alpha > 0$. Then*

$$\Psi_2 \lesssim \frac{d^{1 \vee \frac{p}{2}}}{\alpha n} R.$$

*Proof.* By the triangle inequality,

$$\left| \sum_{i \in G \setminus J} (S_{ii}^p - 1)\Lambda_{ii}^p (|(f(Ax) - v)_i|^p - |(\bar{f}(A\bar{x}) - v)_i|^p) \right|$$

$$\leq \sum_{i \in G \setminus J} (S_{ii}^p + 1)\Lambda_{ii}^p \left| |(f(Ax) - v)_i|^p - |(\bar{f}(A\bar{x}) - v)_i|^p \right|$$

It holds for $i \in G$ and $x \in T$ that

$$|\Lambda_{ii}(f(Ax) - v)_i| \leq |\Lambda_{ii}(f(Ax) - \bar{f}(A\bar{x}))_i| + |\Lambda_{ii}(\bar{f}(A\bar{x}) - v)_i|$$

$$\leq \left( 4 + \frac{1}{\epsilon} \right) d^{\frac{1}{2} - \frac{1}{2\vee p}} w_i^{\frac{1}{p}} R^{\frac{1}{p}}, \qquad \text{(by Lemma A.2)}$$

When one further has $i \notin J$,

$$|\Lambda_{ii}(f(Ax) - v)_i|^p \lesssim \frac{d^{\frac{p}{2} \vee 1 - 1}}{\epsilon^p} w_i R \lesssim \frac{d^{1 \vee \frac{p}{2}}}{n^2} R.$$

Since $S_{ii}^p \leq \frac{1}{\alpha}$, we have

$$
\sup_{(f,x) \in T} \left| \sum_{i \in G \setminus J} (S_{ii}^p - 1) \Lambda_{ii}^p \left( |(f(Ax) - v)_i|^p - |(\bar{f}(A\bar{x}) - v)_i|^p \right) \right|
$$
$$
\lesssim \sum_{i \in G \setminus J} \left( \frac{1}{\alpha} + 1 \right) \frac{d^{1 \vee \frac{p}{2}}}{n^2} R
$$
$$
\lesssim \frac{d^{1 \vee \frac{p}{2}}}{\alpha n} R. \qquad \qquad \square
$$

Next we bound $\Psi_3$, which is the core technical difficulty. Recall that

$$\Psi_3 = \sup_{(f,x) \in T} \left| \sum_{i \in J} (S_{ii}^p - 1) \Lambda_{ii}^p \left( |(f(Ax) - v)_i|^p - |(\bar{f}(A\bar{x}) - v)_i|^p \right) \right|.$$

By the symmetrization trick, we have

$$\mathbb{E}_S \Psi_3^\ell \leq 2^\ell \mathbb{E}_{\xi,S} \left( \sup_{(f,x) \in T} \left| \sum_{i \in J} \xi_i \cdot S_{ii}^p \Lambda_{ii}^p (|(f(Ax) - v)_i|^p - |(\bar{f}(A\bar{x}) - v)_i|^p) \right| \right)^\ell. \tag{11}$$

where $\xi$ has the same dimension as $J$ and whose entries are independent Rademacher random variables.

For every $x \in \mathbb{R}^d$ and $f \in \mathsf{Lip}_1$, we define $Z(f,x) \in \mathbb{R}^n$ coordinatewise as

$$Z_i(f,x) := \Lambda_{ii}^p |(f(Ax) - v)_i|^p.$$

Let $I \subseteq J$ be the set of indices $i$ such that $S_{ii}^p = \frac{1}{\alpha}$. We can then write (11) as

$$\mathbb{E}_S \Psi_3^\ell \leq 2^\ell \mathbb{E}_{\xi,S} \left( \sup_{(f,x) \in T} \left| \frac{1}{\alpha} \langle \xi_I, Z_I(f,x) \rangle - \langle \xi_I, Z_I(\bar{f}, \bar{x}) \rangle \right| \right)^\ell. \tag{12}$$

Next we condition on $S$. Define for $(f_1, x_1), (f_2, x_2) \in \mathsf{Lip}_1 \times \mathbb{R}^d$ the following pseudometric.

$$
D_2((f_1, x_1), (f_2, x_2)) := \left( \sum_{i \in I} |Z_i(f_1, x_1) - Z_i(f_2, x_2)|^2 \right)^{\frac{1}{2}}
$$
$$
= \left( \sum_{i \in I} \left| |\Lambda_{ii}(f_1(Ax_1) - v)_i|^p - |\Lambda_{ii}(f_2(Ax_2) - v)_i|^p \right|^2 \right)^{\frac{1}{2}}.
$$

Since $(\bar{f}, \bar{x}) \in T$, the $\ell$-th moment on the right-hand side of (12) can be upper bounded using Dudley's integral (7) as

$$\mathbb{E}_\xi \left( \sup_{(f,x) \in T} |\langle \xi_I, Z_I(f,x) \rangle - \langle \xi_I, Z_I(\bar{f}, \bar{x}) \rangle| \right)^\ell \leq C^\ell \left( \int_0^\infty \sqrt{\ln \mathcal{N}(T, D_2, \epsilon)} d\epsilon + \sqrt{\ell} \operatorname{diam}(T, D_2) \right)^\ell \tag{13}$$

for some absolute constant $C > 0$. We shall bound the Dudley's integral and the diameter.

## C. Upper Bounding $D_2$

We shall upper bound $D_2$ on $T$ using simpler pseudometrics. Define for $f_1, f_2 \in \mathsf{Lip}_1$ and $x, x' \in \mathbb{R}^d$ the following pseudometrics.

$$d_T(f_1, f_2) := \sup_{x \in \pi_2(T)} \|f_1(Ax) - f_2(Ax)\|_{I, \infty}$$

$$\rho_T(x_1, x_2) := \sup_{f \in \pi_1(T)} \|f(Ax_1) - f(Ax_2)\|_{I, \infty}$$

**Lemma C.1.** *Let* $\{(\bar{f}, \bar{x})\} \subseteq T \subseteq \{(f, x) \in \mathsf{Lip}_1 \times \mathbb{R}^d : \|\Lambda Ax\|_p^p \le R \text{ and } \|\Lambda(\bar{f}(A\bar{x}) - v)\|_p^p \le F\}$. *If* $I \subseteq J$ *satisfies*

$$\|\Lambda(\bar{f}(A\bar{x}) - v)\|_{I,p}^p \lesssim \alpha \|\Lambda(\bar{f}(A\bar{x}) - v)\|_p^p \quad and \quad \sup_{(f,x) \in T} \|\Lambda(f(Ax) - \bar{f}(A\bar{x}))\|_{I,p}^p \lesssim \alpha F,$$

*then*

$$D_2((f_1, x_1), (f_2, x_2)) \lesssim (\alpha F)^\theta \left( (d_T(f_1, f_2))^\phi + (\rho_T(x_1, x_2))^\phi \right),$$

*where*

$$\theta = \left(1 - \frac{1}{p}\right) \vee \frac{1}{2}, \quad \phi = \frac{p}{2} \wedge 1.$$

*Proof.* Recall that

$$D_2((f_1, x_1), (f_2, x_2))^2 = \sum_{i \in I} \left| |\Lambda_{ii}(f_1(Ax_1) - v)_i|^p - |\Lambda_{ii}(f_2(Ax_2) - v)_i|^p \right|^2.$$

For notational convenience, we write $u_1 = f_1(Ax_1)$, $u_2 = f_2(Ax_2)$ and $\bar{u} = \bar{f}(A\bar{x})$. Using the fact that $||a|^p - |b|^p| \le p|a - b|(|a|^{p-1} + |b|^{p-1})$, we have

$$|\Lambda_{ii}(u_1 - v)_i|^p - |\Lambda_{ii}(u_2 - v)_i|^p \le p|\Lambda_{ii}(u_1 - u_2)_i| \left( |\Lambda_{ii}(u_1 - v)_i|^{p-1} + |\Lambda_{ii}(u_2 - v)_i|^{p-1} \right).$$

It then follows that

$$\sum_{i \in I} ||\Lambda_{ii}(u_1 - v)_i|^p - |\Lambda_{ii}(u_2 - v)_i|^p|^2 \lesssim \sum_{i \in I} |\Lambda_{ii}(u_1 - u_2)_i|^2 \left( |\Lambda_{ii}(u_1 - v)_i|^{2p-2} + |\Lambda_{ii}(u_2 - v)_i|^{2p-2} \right).$$

We shall proceed in two ways, depending on the value of $p$.

**Case 1.** $1 \le p \le 2$. We begin with

$$\sum_{i \in I} |\Lambda_{ii}(u_1 - u_2)_i|^2 \left( |\Lambda_{ii}(u_1 - v)_i|^{2p-2} + |\Lambda_{ii}(u_2 - v)_i|^{2p-2} \right)$$

$$\le \left( \max_{i \in I} |\Lambda_{ii}(u_1 - u_2)_i|^p \right) \sum_{i \in I} |\Lambda_{ii}(u_1 - u_2)_i|^{2-p} \left( |\Lambda_{ii}(u_1 - v)_i|^{2p-2} + |\Lambda_{ii}(u_2 - v)_i|^{2p-2} \right).$$

The sum can then be upper bounded as

$$\sum_{i \in I} |\Lambda_{ii}(u_1 - u_2)_i|^{2-p} \left( |\Lambda_{ii}(u_1 - v)_i|^{2p-2} + |\Lambda_{ii}(u_2 - v)_i|^{2p-2} \right)$$

$$\lesssim \sum_{i \in I} \Lambda_{ii}^p |(u_1 - u_2)_i|^{2-p} \max \left\{ |(u_1 - v)_i|^{2p-2}, |(u_2 - v)_i|^{2p-2} \right\}$$

$$\lesssim \left( \sum_{i \in I} \Lambda_{ii}^p |(u_1 - u_2)_i|^p \right)^{\frac{2-p}{p}} \left( \sum_{i \in I} \Lambda_{ii}^p \max \left\{ |(u_1 - v)_i|^p, |(u_2 - v)_i|^p \right\} \right)^{\frac{2p-2}{p}}$$

$$\le \|\Lambda(u_1 - u_2)\|_{I,p}^{2-p} \left( \|\Lambda(u_1 - v)\|_{I,p}^p + \|\Lambda(u_2 - v)\|_{I,p}^p \right)^{\frac{2p-2}{p}}$$

Since $u_j - v = u_j - \bar{u} + \bar{u} - v$ ($j = 1, 2$), by the triangle inequality,

$$\|\Lambda(u_j - v)\|_{I,p} \leq \|\Lambda(u_j - \bar{u})\|_{I,p} + \|\Lambda(\bar{u} - v)\|_{I,p}$$
$$\lesssim (\alpha F)^{\frac{1}{p}} + \left(\alpha\|\Lambda(\bar{u} - v)\|_p^p\right)^{\frac{1}{p}}$$
$$\lesssim (\alpha F)^{\frac{1}{p}}.$$

Moreover,

$$\|\Lambda(u_1 - u_2)\|_{I,p} \leq \|\Lambda(u_1 - \bar{u})\|_{I,p} + \|\Lambda(u_2 - \bar{u})\|_{I,p} \lesssim (\alpha F)^{\frac{1}{p}}.$$

Putting everything together, we have

$$D_2((f_1, x_1), (f_2, x_2))^2 \lesssim \|\Lambda(u_1 - u_2)\|_{I,\infty}^p (\alpha F)^{\frac{2-p}{p}} (\alpha F)^{\frac{2p-2}{p}} \leq \alpha F \|\Lambda(u_1 - u_2)\|_{I,\infty}^p.$$

**Case 2.** $p > 2$. Note that for any vector $z$, we have $|z_i|^{2p-2} \leq |z_i|^p \|z\|_\infty^{p-2}$. Hence,

$$\sum_{i \in I} |\Lambda_{ii}(u_1 - u_2)_i|^2 \left(|\Lambda_{ii}(u_1 - v)_i|^{2p-2} + |\Lambda_{ii}(u_2 - v)_i|^{2p-2}\right)$$
$$\leq \sum_{i \in I} \|\Lambda(u_1 - u_2)\|_{I,\infty}^2 \left(|\Lambda_{ii}(u_1 - v)_i|^p \|\Lambda(u_1 - v)\|_{I,\infty}^{p-2} + |\Lambda_{ii}(u_2 - v)_i|^p \|\Lambda(u_2 - v)\|_{I,\infty}^{p-2}\right)$$
$$= \|\Lambda(u_1 - u_2)\|_{I,\infty}^2 \left(\|\Lambda(u_1 - v)\|_{I,p}^p \|\Lambda(u_1 - v)\|_{I,\infty}^{p-2} + \|\Lambda(u_2 - v)\|_{I,p}^p \|\Lambda(u_2 - v)\|_{I,\infty}^{p-2}\right)$$
$$\leq (\alpha F)\|\Lambda(u_1 - u_2)\|_{I,\infty}^2 \left(\|\Lambda(u_1 - v)\|_{I,\infty}^{p-2} + \|\Lambda(u_2 - v)\|_{I,\infty}^{p-2}\right)$$
$$\leq (\alpha F)\|\Lambda(u_1 - u_2)\|_{I,\infty}^2 \left(\|\Lambda(u_1 - v)\|_{I,p}^{p-2} + \|\Lambda(u_2 - v)\|_{I,p}^{p-2}\right).$$

Recall that $\|\Lambda(u_1 - v)\|_{I,p} \lesssim (\alpha F)^{\frac{1}{p}}$ and $\|\Lambda(u_2 - v)\|_{I,p} \lesssim (\alpha F)^{\frac{1}{p}}$. It then follows that

$$D_2((f_1, x_1), (f_2, x_2))^2 \lesssim (\alpha F)(\alpha F)^{\frac{p-2}{p}} \|\Lambda(u_1 - u_2)\|_{I,\infty}^2 \leq (\alpha F)^{2-\frac{2}{p}} \|\Lambda(u_1 - u_2)\|_{I,\infty}^2.$$

Combining both cases, we obtain that

$$D_2((f_1, x_1), (f_2, x_2)) \lesssim (\alpha F)^{\frac{1}{2} \vee (1 - \frac{1}{p})} \|f_1(Ax_1) - f_2(Ax_2)\|_{I,\infty}^{\frac{p}{2} \wedge 1}. \tag{14}$$

By the triangle inequality,

$$\|f_1(Ax_1) - f_2(Ax_2)\|_{I,\infty} \leq \|f_1(Ax_1) - f_2(Ax_1)\|_{I,\infty} + \|f_2(Ax_1) - f_2(Ax_2)\|_{I,\infty}. \tag{15}$$

Plugging (15) into (14) yields immediately that

$$D_2((f_1, x_1), (f_2, x_2)) \lesssim (\alpha F)^{\frac{1}{2} \vee (1 - \frac{1}{p})} \left[\sup_{x \in \pi_2(T)} \|f_1(Ax) - f_2(Ax)\|_{I,\infty}^{\frac{p}{2} \wedge 1} + \sup_{f \in \pi_1(T)} \|f(Ax_1) - f(Ax_2)\|_{I,\infty}^{\frac{p}{2} \wedge 1}\right]$$

as desired. □

**Lemma C.2.** *Under the assumptions of Lemma C.1, it holds that*

$$\int_0^\infty \sqrt{\ln \mathcal{N}(T, D_2, \epsilon)} d\epsilon \lesssim (\alpha F)^\theta \left[\int_0^\infty \sqrt{\ln \mathcal{N}(\pi_1(T), d_T^\phi, \epsilon)} d\epsilon + \int_0^\infty \sqrt{\ln \mathcal{N}(\pi_2(T), \rho_T^\phi, \epsilon)} d\epsilon\right] \tag{16}$$

*and*

$$\mathrm{diam}(T, D_2) \leq (\alpha F)^\theta \left[\mathrm{diam}(\pi_1(T), d_T^\phi) + \mathrm{diam}(\pi_2(T), \rho_T^\phi)\right].$$

*Proof.* Lemma C.1 implies that

$$\mathrm{diam}(T, D_2) \leq (\alpha F)^{\theta} \left[ \mathrm{diam}(\pi_1(T), d_T^{\phi}) + \mathrm{diam}(\pi_2(T), \rho_T^{\phi}) \right]$$

and

$$\mathcal{N}(T, D_2, \epsilon) \leq \mathcal{N}\left(\mathsf{Lip}_1, (\alpha F)^{\theta} d_T^{\phi}, \frac{\epsilon}{2}\right) \cdot \mathcal{N}\left(T, (\alpha F)^{\theta} \rho_T^{\phi}, \frac{\epsilon}{2}\right).$$

It follows that

$$\sqrt{\ln \mathcal{N}(T, D_2, \epsilon)} \leq \sqrt{\ln \mathcal{N}\left(\mathsf{Lip}_1, (\alpha F)^{\theta} d_T^{\phi}, \frac{\epsilon}{2}\right)} + \sqrt{\ln \mathcal{N}\left(T, (\alpha F)^{\theta} \rho_T^{\phi}, \frac{\epsilon}{2}\right)}$$

and

$$\int_0^{\infty} \sqrt{\ln \mathcal{N}(T, D_2, \epsilon)} d\epsilon \lesssim \int_0^{\infty} \sqrt{\ln \mathcal{N}\left(\pi_1(T), (\alpha F)^{\theta} d_T^{\phi}, \frac{\epsilon}{2}\right)} d\epsilon + \int_0^{\infty} \sqrt{\ln \mathcal{N}\left(\pi_2(T), (\alpha F)^{\theta} \rho_T^{\phi}, \frac{\epsilon}{2}\right)} d\epsilon$$

$$\leq 2(\alpha F)^{\theta} \left( \int_0^{\infty} \sqrt{\ln \mathcal{N}(\pi_1(T), d_T^{\phi}, \epsilon)} d\epsilon + \int_0^{\infty} \sqrt{\ln \mathcal{N}(\pi_2(T), \rho_T^{\phi}, \epsilon)} d\epsilon \right). \qquad \square$$

We shall bound the two Dudley's integrals in the right-hand side of (16) separately. The second integral regarding $\pi_2(T)$ was computed in (Li & Tai, 2025) for the problem with a known $f$. We summarize the result in the following lemma.

**Lemma C.3** ((Li & Tai, 2025)). *Under the assumptions of Lemma C.1, it holds that*

$$(\alpha F)^{\theta} \int_0^{\infty} \sqrt{\ln \mathcal{N}\left(\pi_2(T), \rho_T^{\phi}, \epsilon\right)} d\epsilon \lesssim \alpha \Gamma (\ln^{\frac{5}{4}} d) \sqrt{\ln \frac{n}{\epsilon d}}, \quad (\alpha F)^{\theta} \mathrm{diam}(\pi_2(T), \rho_T^{\phi}) \lesssim \alpha \Gamma, \qquad (17)$$

*where $\Gamma$ is defined as in* (6).

## D. Bounds Pertaining to $\mathsf{Lip}_1$

It remains to upper bound the first integral regarding $\mathsf{Lip}_1$. First, we prove an auxiliary result that bounds the covering number of $\mathsf{Lip}_1$ under $\|\cdot\|$, where $\|\cdot\|$ is the supremum of a collection of $L^{\infty}$-norms on different intervals, by the covering number under the $L^{\infty}$-norm on $[-1, 1]$.

**Theorem D.1.** *Suppose that each $i$ in a finite index set $I$ corresponds to a bound threshold $M_i > 0$ and a weight $\lambda_i \in (0, 1)$. Consider the norm $\|\cdot\|$ on $\mathsf{Lip}_1$ defined as*

$$\|f\| := \max_{i \in I} \lambda_i \|f\|_{L^{\infty}([-M_i, M_i])}.$$

*Let $M = \max_{i \in I} \lambda_i M_i$ and $\kappa = (\max_{i \in I} M_i)/M$. It holds for $\epsilon \in (0, 1)$ that*

$$\mathcal{N}(\mathsf{Lip}_1, \|\cdot\|, \epsilon) \leq \mathcal{N}\left(\mathsf{Lip}_1, \|\cdot\|_{L^{\infty}([-1,1])}, \frac{\epsilon}{2M(\ln \kappa + 1)}\right).$$

*Proof.* Let $\beta = 1/(1 + \ln \kappa)$. For each $f \in \mathsf{Lip}_1$, define $Tf : [-1, 1] \to \mathbb{R}$ as

$$Tf(x) := \begin{cases} \frac{1}{M} f\left(\frac{M}{\beta} x\right), & |x| \leq \beta \\ f\left(\mathrm{sgn}(x) \cdot \kappa^{\frac{|x|-\beta}{1-\beta}} M\right) / \left(\kappa^{\frac{|x|-\beta}{1-\beta}} M\right), & |x| > \beta \end{cases}$$

This map $T$ is inspired by the discretization of Lipschitz functions in (Gajjar et al., 2024).

We claim that $Tf$ is $L$-Lipschitz for $L = 2(\ln \kappa + 1)$. It is clear that when $x, y \in [-\beta, \beta]$,

$$|Tf(x) - Tf(y)| \leq \frac{1}{\beta}|x - y| = (\ln \kappa + 1)|x - y|.$$

Suppose that $1 \geq x > y \geq \beta$. Note that $\kappa^{\frac{x-\beta}{1-\beta}} = \kappa^x e^{x-1}$, thus

$$
\begin{aligned}
|Tf(x) - Tf(y)| &= \left| \frac{f(\kappa^x e^{x-1} M)}{\kappa^x e^{x-1} M} - \frac{f(\kappa^y e^{y-1} M)}{\kappa^y e^{y-1} M} \right| \\
&= \left| \frac{f(\kappa^x e^{x-1} M)}{\kappa^{x-y} e^{x-y} \kappa^y e^{y-1} M} - \frac{f(\kappa^y e^{y-1} M)}{\kappa^y e^{y-1} M} \right| \\
&= \left| \frac{f(\kappa^x e^{x-1} M) - f(\kappa^y e^{y-1} M)}{\kappa^{x-y} e^{x-y} \kappa^y e^{y-1} M} - \frac{(\kappa^{x-y} e^{x-y} - 1) f(\kappa^y e^{y-1} M)}{\kappa^{x-y} e^{x-y} \kappa^y e^{y-1} M} \right| \\
&\leq \frac{\kappa^x e^{x-1} M - \kappa^y e^{y-1} M}{\kappa^{x-y} e^{x-y} \kappa^y e^{y-1} M} + \frac{\kappa^{x-y} e^{x-y} - 1}{\kappa^{x-y} e^{x-y}} \\
&= 2 \cdot \frac{\kappa^{x-y} e^{x-y} - 1}{\kappa^{x-y} e^{x-y}} \\
&= 2(1 - (\kappa e)^{-(x-y)}) \\
&\leq 2(x-y) \ln(\kappa e).
\end{aligned}
$$

A similar bound can be established on $[-1, -\beta]$. This establishes that $Tf$ is $L$-Lipschitz for $L = 2(\ln \kappa + 1)$.

Next we show that $\|f\| \leq M \|Tf\|_{L^\infty([-1,1])}$. Indeed, fix an $i$ and suppose that $|f(x_i)| = \|f\|_{L^\infty([-M_i, M_i])}$.

**Case 1.** $|x_i| \leq M$. Recall that $\lambda_i \in (0, 1)$, we have

$$
\lambda_i |f(x_i)| \leq |f(x_i)| = M \left| (Tf) \left( \frac{\beta}{M} x_i \right) \right| \leq M \|Tf\|_{L^\infty([-1,1])}.
$$

**Case 2.** $x_i > M$. We have

$$
f(x_i) = x_i \cdot (Tf) \left( \beta \left( 1 + \ln \frac{x_i}{M} \right) \right).
$$

Hence,

$$
\lambda_i |f(x_i)| \leq |\lambda_i x_i| \|Tf\|_{L^\infty([-1,1])} \leq M \|Tf\|_{L^\infty([-1,1])}. \tag{18}
$$

**Case 3.** $x_i < -M$. The argument is similar to that of $x_i > M$ and (18) also holds.

Combining these cases yields that $\|f\| \leq M \|Tf\|_{L^\infty([-1,1])}$.

Therefore, for every $f \in \mathsf{Lip}_1$, there exists $g \in \mathsf{Lip}_1$ such that

$$
\|f\| \leq ML \|g\|_{L^\infty([-1,1])},
$$

which implies that

$$
\mathcal{N}(\mathsf{Lip}_1, \|\cdot\|, \epsilon) \leq \mathcal{N} \left( \mathsf{Lip}_1, \|\cdot\|_{L^\infty([-1,1])}, \frac{\epsilon}{2M(\ln \kappa + 1)} \right). \qquad \square
$$

The following entropy bound for Lipschitz functions is standard; see, e.g., Lemma 4.5.18 of (Talagrand, 2021).

**Lemma D.2** ((Talagrand, 2021)). *It holds for $\epsilon > 0$ that*

$$
\ln \mathcal{N}(\mathsf{Lip}_1, \|\cdot\|_{L^\infty([-1,1])}, \epsilon) \lesssim \frac{1}{\epsilon}.
$$

Now we are ready to upper bound the Dudley's integral for $\mathsf{Lip}_1$ in (16).

**Lemma D.3.** *Suppose that $p > 1$ is a constant. Under the assumptions of Lemma C.1, it holds that*

$$
\int_0^\infty \sqrt{\ln \mathcal{N}(\pi_1(T), d_T^\phi, \epsilon)} \, d\epsilon \lesssim \left( \left( \frac{d}{n} \right)^{\frac{1}{p}} d^{\frac{1}{2} - \frac{1}{2 \vee p}} R^{\frac{1}{p}} \ln \frac{n}{d} \right)^{\frac{p}{2} \wedge 1}
$$

*and*

$$
\mathrm{diam}(\pi_1(T), d_T^\phi) \lesssim \left( \left( \frac{d}{n} \right)^{\frac{1}{p}} d^{\frac{1}{2} - \frac{1}{2 \vee p}} R^{\frac{1}{p}} \ln \frac{n}{d} \right)^{\frac{p}{2} \wedge 1}
$$

*Proof.* It follows from the definition of $d_T$ that

$$
(d_T(f_1, f_2))^\phi = \sup_{x \in \pi_2(T)} \|\Lambda(f_1(Ax) - f_2(Ax))\|_{I,\infty}^\phi
$$

$$
= \left( \max_{i \in I} \Lambda_{ii} \sup_{x \in \pi_2(T)} |(f_1(Ax) - f_2(Ax))_i| \right)^\phi
$$

$$
\leq \|f_1 - f_2\|^\phi,
$$

where

$$
\|f\| = \max_{i \in I} \Lambda_{ii} \|f\|_{L^\infty([-M_i, M_i])}, \quad M_i = \sup_{x \in \pi_2(T)} |(Ax)_i|.
$$

We know from Lemma A.1 that

$$
\max_{i \in I} \Lambda_{ii} M_i \lesssim \left( \frac{d}{n} \right)^{\frac{1}{p}} d^{\frac{1}{2} - \frac{1}{2 \vee p}} R^{\frac{1}{p}} =: M
$$

and from the fact that $\Lambda_{ii} \geq (n/d)^{-1/p}$ that

$$
\kappa = \frac{\max_{i \in I} M_i}{M} \lesssim \max_i \frac{1}{\Lambda_{ii}} \leq \left( \frac{n}{d} \right)^{\frac{1}{p}}.
$$

We apply Theorem D.1 to the cases $p > 2$ and $p \leq 2$ separately below.

**Case 1.** $1 < p \leq 2$. Let $\Delta$ be such that $\Delta^{\frac{2}{p}} = 2M \ln \kappa$.

$$
\int_0^\infty \sqrt{\ln \mathcal{N}(\pi_1(T), d_T^\phi, \epsilon)} d\epsilon \leq \int_0^\infty \sqrt{\ln \mathcal{N}(\mathsf{Lip}_1, \|\cdot\|^\phi, \epsilon)} d\epsilon
$$

$$
= \int_0^\infty \sqrt{\ln \mathcal{N}\left(\mathsf{Lip}_1, \|\cdot\|^{\frac{p}{2}}, \epsilon\right)} d\epsilon
$$

$$
= \int_0^\infty \sqrt{\ln \mathcal{N}\left(\mathsf{Lip}_1, \|\cdot\|, \epsilon^{\frac{2}{p}}\right)} d\epsilon
$$

$$
\leq \int_0^\infty \sqrt{\ln \mathcal{N}\left(\mathsf{Lip}_1, \|\cdot\|_{L^\infty([-1,1])}, \frac{\epsilon^{\frac{2}{p}}}{2M \ln \kappa}\right)} d\epsilon
$$

$$
\leq \int_0^\Delta \frac{\sqrt{2M \ln \kappa}}{\epsilon^{\frac{1}{p}}} d\epsilon
$$

$$
\asymp_p \sqrt{M \ln \kappa} \cdot \Delta^{1 - \frac{1}{p}}
$$

$$
= \Delta^{\frac{1}{p}} \cdot \Delta^{1 - \frac{1}{p}}
$$

$$
= \Delta
$$

$$
= (2M \ln \kappa)^{\frac{p}{2}}.
$$

**Case 2.** $p > 2$. Let $\Delta = 2M \ln \kappa$.

$$
\int_0^\infty \sqrt{\ln \mathcal{N}(\pi_1(T), d_T^\phi, \epsilon)} d\epsilon \leq \int_0^\infty \sqrt{\ln \mathcal{N}(\mathsf{Lip}_1, \|\cdot\|, \epsilon)} d\epsilon
$$

$$
\lesssim \int_0^\infty \sqrt{\ln \mathcal{N}\left(\mathsf{Lip}_1, \|\cdot\|_{L^\infty([-1,1])}, \frac{\epsilon}{2M \ln \kappa}\right)} d\epsilon
$$

$$
\lesssim \int_0^\Delta \frac{\sqrt{2M \ln \kappa}}{\epsilon^{\frac{1}{2}}} d\epsilon
$$

$$
\asymp \sqrt{2M \ln \kappa} \Delta^{\frac{1}{2}}
$$

$$
= \Delta^{\frac{1}{2}} \Delta^{\frac{1}{2}}
$$

$$
= \Delta
$$

$$
= 2M \ln \kappa
$$

Combining both cases leads to

$$\int_0^\infty \sqrt{\ln \mathcal{N}(\pi_1(T), d_T^\phi, \epsilon)} d\epsilon \lesssim (M \ln \kappa)^\phi,$$

which is exactly the first claimed result.

Moreover, we know from the proof above that

$$\mathrm{diam}(\pi_1(T), d_T^\phi) \le (\mathrm{diam}(\mathsf{Lip}_1, \|\cdot\|))^\phi \lesssim \left(M \ln \kappa \cdot \mathrm{diam}(\mathsf{Lip}_1, \|\cdot\|_{L^\infty([-1,1])})\right)^\phi \lesssim (M \ln \kappa)^\phi.$$

This is exactly the second claimed result. □

**Corollary D.4.** *Suppose that $p > 1$ is a constant. Under the assumptions of Lemma C.1, it holds that*

$$(\alpha F)^\theta \int_0^\infty \sqrt{\ln \mathcal{N}(\pi_1(T), d_T^\phi, \epsilon)} d\epsilon \lesssim \alpha \Gamma \ln \frac{n}{d}$$

*and*

$$(\alpha F)^\theta \mathrm{diam}(\pi_1(T), d_T^\phi) \lesssim \alpha \Gamma \ln \frac{n}{d}.$$

*where $\Gamma$ is defined as in (6).*

## E. Main Error Bound for Upper Bound Analysis

Based on the preceding sections, we now upper bound $\Psi_3$ as follows.

**Lemma E.1.** *Suppose that $p > 1$ is a constant. Under the assumptions of C.1, it holds with probability at least $1 - \delta$ that*

$$\Psi_3 \lesssim \Gamma \ln \frac{n}{\epsilon d} \left( \ln^{\frac{5}{4}} d + \sqrt{\ln \frac{1}{\delta}} \right),$$

*where $\Gamma$ is as defined in (6).*

*Proof.* For ease of notations again we let $M := \varphi R^{\frac{1}{p}} = \left(\frac{d}{n}\right)^{\frac{1}{p}} d^{\frac{1}{2} - \frac{1}{2 \vee p}} R^{\frac{1}{p}}$ and $\kappa := R^{\frac{1}{p}}/M \asymp \mathrm{poly}(\frac{n}{d})$.

Combining Lemma C.3 and Corollary D.4 with Lemma C.2, we obtain that

$$\int_0^\infty \sqrt{\ln \mathcal{N}(T, D_2, \epsilon)} d\epsilon \lesssim \alpha \Gamma \left( \ln \frac{n}{d} + (\ln^{\frac{5}{4}} d) \sqrt{\ln \frac{n}{\epsilon d}} \right) \lesssim \alpha \Gamma (\ln^{\frac{5}{4}} d) \ln \frac{n}{\epsilon d}. \tag{19}$$

and

$$\mathrm{diam}(T, D_2) \lesssim \alpha \Gamma \ln \frac{n}{d}. \tag{20}$$

Combining (12), (13) and (19), (20), we obtain that

$$\left( \mathbb{E}_S \Psi_3^\ell \right)^{\frac{1}{\ell}} \lesssim \Gamma \ln \frac{n}{\epsilon d} \left( \ln^{\frac{5}{4}} d + \sqrt{\ell} \right).$$

Taking $\ell = \log(1/\delta)$ and applying Markov's inequality leads to the claimed result. □

All the results so far give rise to the following main theorem.

**Theorem 3.1.** *Let $p > 1$ be a constant, $A \in \mathbb{R}^{n \times d}$, $\epsilon \in (0, 1)$ be sufficiently small, and $\Lambda \in \mathbb{R}^{n \times n}$ be a diagonal matrix with entries $\Lambda_{ii} > 0$ such that $w_i(\Lambda A) \lesssim d/n$ for all $i \in [n]$. Let $v \in \mathbb{R}^n$ be a fixed vector, $(\bar{f}, \bar{x}) \in \mathsf{Lip}_1 \times \mathbb{R}^d$ be a fixed reference point, and $V := \|\Lambda(\bar{f}(A\bar{x}) - v)\|_p^p$.*

*Let $R \ge \|\Lambda A\bar{x}\|_p^p$ and $F \ge V$ be fixed upper bounds. Suppose the set $T$ satisfies*

$$\{(\bar{f}, \bar{x})\} \subseteq T \subseteq \{(f, x) \in \mathsf{Lip}_1 \times \mathbb{R}^d : \|\Lambda A x\|_p^p \le R \text{ and } \|\Lambda(f(Ax) - \bar{f}(A\bar{x}))\|_p^p \le F\}.$$

*Let $\alpha \in [0,1]$ and $S$ be an $n \times n$ random diagonal matrix with i.i.d. $\alpha^{-1/p}\operatorname{Ber}(\alpha)$ entries. Conditioned on the event that*

$$\|S\Lambda(\bar{f}(A\bar{x}) - v)\|_p^p \lesssim V \quad \text{and} \quad \sup_{(f,x)\in T} \|S\Lambda(f(Ax) - \bar{f}(A\bar{x}))\|_p^p \leq F,$$

*it holds with probability at least $1 - \delta$ that*

$$\sup_{(f,x)\in T} \left| \left(\|S\Lambda(f(Ax) - v)\|_p^p - \|\Lambda(f(Ax) - v)\|_p^p\right) - \left(\|S\Lambda(\bar{f}(A\bar{x}) - v)\|_p^p - \|\Lambda(\bar{f}(A\bar{x}) - v)\|_p^p\right) \right|$$

$$\leq C\left(\epsilon V + \frac{d^{1\vee\frac{p}{2}}}{\alpha n}R + \Gamma \ln \frac{n}{\epsilon d}\left(\ln^{\frac{5}{4}} d + \sqrt{\ln\frac{1}{\delta}}\right)\right),$$

*where $C$ depends only on $p$, and $\Gamma$ is defined as*

$$\Gamma := \begin{cases} d^{\frac{1}{2}}(\alpha n)^{-\frac{1}{2}}F^{\frac{1}{2}}R^{\frac{1}{2}}, & 1 \leq p \leq 2 \\ d^{\frac{1}{2}}(\alpha n)^{-\frac{1}{p}}F^{1-\frac{1}{p}}R^{\frac{1}{p}}, & p > 2. \end{cases} \tag{6}$$

With the preceding theorem established, we immediately obtain the following two corollaries.

**Corollary E.2.** *Suppose $\alpha \gtrsim \frac{d^{1\vee\frac{p}{2}}}{n}$. When conditioned on the event that $\|S\Lambda Ax\|_p^p \lesssim \|\Lambda Ax\|_p^p$ for all $x \in \mathbb{R}^d$, it holds with probability at least $1 - \delta$ that*

$$\sup_{(f,x)\in T} \left| \|S\Lambda(f(Ax) - \bar{f}(A\bar{x}))\|_p^p - \|\Lambda(f(Ax) - \bar{f}(A\bar{x}))\|_p^p \right|$$

$$\lesssim \frac{d^{\frac{1}{2}}}{(\alpha n)^{\frac{1}{2\vee p}}}R \ln \frac{n}{\epsilon d}\left(\ln^{\frac{5}{4}} d + \sqrt{\ln\frac{1}{\delta}}\right).$$

*Proof.* In the above theorem take $v = \bar{f}(A\bar{x})$ and $\epsilon$ to be a constant. Then $\|S\Lambda(\bar{f}(A\bar{x}) - v)\|_p^p = V = 0$. We have the following upper bound:

$$\|S\Lambda(f(Ax) - \bar{f}(A\bar{x}))\|_p^p \leq 2^{p-1}(\|S\Lambda(f(Ax) - \bar{f}(Ax))\|_p^p + \|S\Lambda(\bar{f}(Ax) - \bar{f}(A\bar{x}))\|_p^p)$$

$$\leq 2^{p-1}(2^p\|S\Lambda Ax\|_p^p + \|S\Lambda(Ax - A\bar{x})\|_p^p)$$

$$\leq 2^{p-1}(2^p C_1\|\Lambda Ax\|_p^p + C_1\|\Lambda(Ax - A\bar{x})\|_p^p)$$

$$\leq 2^{p-1}(2^p C_1\|\Lambda Ax\|_p^p + C_1 2^p(\|\Lambda Ax\|_p^p + \|\Lambda A\bar{x}\|_p^p))$$

$$\leq 4^p C_1 R =: F$$

and thus the result follows from Theorem 3.1. $\square$

**Corollary E.3.** *Suppose that $\alpha \gtrsim \frac{d^{1\vee\frac{p}{2}}}{n\epsilon}$ and $F \gtrsim \epsilon R$. When conditioned on the event that*

$$\|S\Lambda(\bar{f}(A\bar{x}) - b)\|_p^p \lesssim \|\Lambda(\bar{f}(A\bar{x}) - b)\|_p^p \quad \text{and} \quad \sup_{(f,x)\in T} \|S\Lambda(f(Ax) - \bar{f}(A\bar{x}))\|_p^p \leq F,$$

*it holds with probability at least $1 - \delta$ that*

$$\sup_{(f,x)\in T} \left| \left(\|S\Lambda(f(Ax) - b)\|_p^p - \|\Lambda(f(Ax) - b)\|_p^p\right) - \left(\|S\Lambda(\bar{f}(A\bar{x}) - b)\|_p^p - \|\Lambda(\bar{f}(A\bar{x}) - b)\|_p^p\right) \right|$$

$$\lesssim \Gamma \cdot \ln \frac{n}{\epsilon d}\left(\ln^{\frac{5}{4}} d + \sqrt{\log\frac{1}{\delta}}\right)$$

*Proof.* In Theorem 3.1 take $v = b$. We have $V = \|\Lambda(\bar{f}(A\bar{x}) - b)\|_p^p$ and the result follows, noticing that the last term in the error bound of Theorem 3.1 is the dominating term. $\square$

# F. $(1+\epsilon)$-Approximation

**Theorem F.1.** *Let $p \geq 1$ be a constant, $A \in \mathbb{R}^{n \times d}$, $f, \bar{f} \in \mathsf{Lip}_1$, $\epsilon \in (0,1)$ be sufficiently small, and $\Lambda$ be an $n \times n$ diagonal matrix satisfying $\Lambda_{ii} > 0$ and $w_i(\Lambda A) \lesssim d/n$ for all $1 \leq i \leq n$. Let $S$ be $n \times n$ random diagonal matrix in which the diagonal entries are i.i.d. $\alpha^{-\frac{1}{p}} Ber(\alpha)$ variables, where $\alpha \gtrsim \frac{d^{\frac{p}{2} \vee 1}}{n \epsilon^{p \vee 2}} \operatorname{poly} \log \frac{n}{\epsilon}$.*

*If $\hat{x}, \bar{x} \in \mathbb{R}^d$ and $\hat{f}, \bar{f} \in \mathsf{Lip}_1$ satisfy*

$$(\hat{f}, \hat{x}) = \underset{x \in \mathbb{R}^d,\, f \in \mathsf{Lip}_1}{\arg \min} \|S\Lambda(f(Ax) - b)\|_p^p + \epsilon \|\Lambda Ax\|_p^p,$$

*and*

$$\|\Lambda(\bar{f}(A\bar{x}) - b)\|_p^p - \|\Lambda(\hat{f}(A\hat{x}) - b)\|_p^p \lesssim \epsilon(\|\Lambda(\bar{f}(A\bar{x}) - b)\|_p^p + \epsilon\|\Lambda A\bar{x}\|_p^p),$$

*then with probability at least $0.9$,*

$$\left| \left( \|S\Lambda(\hat{f}(A\hat{x}) - b)\|_p^p - \|\Lambda(\hat{f}(A\hat{x}) - b)\|_p^p \right) - \left( \|S\Lambda(\bar{f}(A\bar{x}) - b)\|_p^p - \|\Lambda(\bar{f}(A\bar{x}) - b)\|_p^p \right) \right|$$
$$\leq \epsilon \left( \|\Lambda(\bar{f}(A\bar{x}) - b)\|_p^p + \|\Lambda A\bar{x}\|_p^p \right).$$

*Proof.* We assume that $p > 1$. The proof for the case of $p = 1$ is deferred to Appendix G.

By the optimality of $(\hat{f}, \hat{x})$, we have

$$\|S\Lambda(\hat{f}(A\hat{x}) - b)\|_p^p + \epsilon \|\Lambda A\hat{x}\|_p^p \leq \|S\Lambda(\bar{f}(A\bar{x}) - b)\|_p^p + \epsilon \|\Lambda A\bar{x}\|_p^p$$

By Markov's inequality, with probability at least $0.99$,

$$\|S\Lambda(\bar{f}(A\bar{x}) - b)\|_p^p \leq 100 \|\Lambda(\bar{f}(A\bar{x}) - b)\|_p^p.$$

We conditioned on this event in the remainder of the proof. This implies that

$$\|\Lambda A\hat{x}\|_p^p \leq \frac{1}{\epsilon} \|S\Lambda(\bar{f}(A\bar{x}) - b)\|_p^p + \|\Lambda A\bar{x}\|_p^p$$
$$\leq \frac{100}{\epsilon} \|\Lambda(\bar{f}(A\bar{x}) - b)\|_p^p + \|\Lambda A\bar{x}\|_p^p$$
$$:= R_0.$$

Throughout the rest of the proof, we assume that $\alpha \gtrsim \frac{d^{1 \vee \frac{p}{2}}}{n \epsilon^{p \vee 2}} \operatorname{poly}(\ln n)$ and $\delta \sim 1/\log \log \frac{1}{\epsilon}$ so that the error term in Corollary E.3 can be bounded as

$$\Gamma \ln \frac{n}{\epsilon d} \cdot \left( \operatorname{poly}(\ln d) + \sqrt{\ln \frac{1}{\delta}} \right) \lesssim \epsilon F^\theta R^\beta,$$

where $\beta = \frac{1}{2} \wedge \frac{1}{p}$ and $\theta = (1 - \frac{1}{p}) \vee \frac{1}{2}$. It is obvious that $\theta + \beta = 1$.

**Bounding $F$ in Corollary E.3.** Let $T_{-1} = \{(f, x) \in \mathsf{Lip}_1 \times \mathbb{R}^d : \|\Lambda Ax\|_p^p \leq R_0\}$, where $R_0 = \frac{100}{\epsilon} \|\Lambda(\bar{f}(A\bar{x}) - b)\|_p^p + \|\Lambda A\bar{x}\|_p^p$.

By Corollary E.2 with our choice of $\alpha$ and $R = R_0$, it holds with probability at least $0.99$ that

$$\sup_{(f,x) \in T_{-1}} \left| \|S\Lambda(f(Ax) - \bar{f}(A\bar{x}))\|_p^p - \|\Lambda(f(Ax) - \bar{f}(A\bar{x}))\|_p^p \right| \leq C_1 \epsilon R_0 \tag{21}$$

where $C_1$ is a constant that depend only on $p$. Below we shall use constants $C_2, C_3, \ldots$ to denote constants that depends only on $p$.

Conditioning on the event in (21), we have

$$\|\Lambda(\hat{f}(A\hat{x}) - \bar{f}(A\bar{x}))\|_p^p$$

$$\leq \|S\Lambda(\hat{f}(A\hat{x}) - \bar{f}(A\bar{x}))\|_p^p + C_1\epsilon R_0$$

$$\leq 2^{p-1}(\|S\Lambda(\hat{f}(A\hat{x}) - b)\|_p^p + \|S\Lambda(\bar{f}(A\bar{x}) - b)\|_p^p) + C_1\epsilon R_0$$

$$\leq 2^{p-1}(\|S\Lambda(\bar{f}(A\bar{x}) - b)\|_p^p + \epsilon\|\Lambda A\bar{x}\|_p^p + \|S\Lambda(\bar{f}(A\bar{x}) - b)\|_p^p) + C_1\epsilon R_0$$

$$= 2^{p-1}(2\|S\Lambda(\bar{f}(A\bar{x}) - b)\|_p^p + \epsilon\|\Lambda A\bar{x}\|_p^p) + C_1\epsilon R_0$$

$$\leq 2^{p-1}(2 \cdot 100\|\Lambda(\bar{f}(A\bar{x}) - b)\|_p^p + \epsilon\|\Lambda A\bar{x}\|_p^p) + C_1\epsilon\left(\frac{100}{\epsilon}\|\Lambda(\bar{f}(A\bar{x}) - b)\|_p^p + \|\Lambda A\bar{x}\|_p^p\right)$$

$$\leq C_2(\|\Lambda(\bar{f}(A\bar{x}) - b)\|_p^p + \epsilon\|\Lambda A\bar{x}\|_p^p)$$

for some large constant $C_2$. Define

$$F_0 := C_2(\|\Lambda(\bar{f}(A\bar{x}) - b)\|_p^p + \epsilon\|\Lambda A\bar{x}\|_p^p).$$

**Defining $T_i$ and $R_i$ in Corollary E.3.** Recall that $R_0 = \frac{100}{\epsilon}\|\Lambda(\bar{f}(A\bar{x}) - b)\|_p^p + \|\Lambda A\bar{x}\|_p^p$.

We shall define $R_i$ based on $R_{i-1}$ ensuring that $R_i \leq R_0$ and that each $R_i$ has the form $X_i\|\Lambda(\bar{f}(A\bar{x}) - b)\|_p^p + Y_i\|\Lambda A\bar{x}\|_p^p$ for some $X_i, Y_i \geq 1$. Furthermore let,

$$T_i = \left\{(f, x) \in \mathsf{Lip}_1 \times \mathbb{R}^d : \|\Lambda Ax\|_p^p \leq R_i \quad \text{and} \quad \|\Lambda(f(Ax) - \bar{f}(A\bar{x}))\|_p^p \leq F_0\right\}$$

so that $T_i \subseteq T_0$.

It is obvious that $R_i \leq R_0 \lesssim \frac{1}{\epsilon}F_0$. We also have

$$\|S\Lambda(\bar{f}(A\bar{x}) - b)\|_p^p \lesssim \|\Lambda(\bar{f}(A\bar{x}) - b)\|_p^p,$$

and by (21) and the fact that $T_i \subseteq T_{-1}$ that

$$\sup_{(f,x)\in T_i} \|S\Lambda(f(Ax) - \bar{f}(A\bar{x}))\|_p^p \leq \sup_{(f,x)\in T_i} \|\Lambda(f(Ax) - \bar{f}(A\bar{x}))\|_p^p + C_1\epsilon R_0 \lesssim F_0.$$

Using our choice of $\alpha$, $R = R_i$ and $F = F_0$, we have by Corollary E.3 that with probability at least $1 - \delta$

$$\sup_{(f,x)\in T_i} \left|(\|S\Lambda(f(Ax) - b)\|_p^p - \|\Lambda(f(Ax) - b)\|_p^p) - (\|S\Lambda(\bar{f}(A\bar{x}) - b)\|_p^p - \|\Lambda(\bar{f}(A\bar{x}) - b)\|_p^p)\right| \leq C_3\epsilon R_i^\beta F_0^\theta \quad (22)$$

for some constant $C_3$.

**Bootstrapping.** We would like to argue that the solution $(\hat{f}, \hat{x}) \in T_i$. For $T_0$ we have

$$\|\Lambda A\hat{x}\|_p^p \leq R_0 \quad \text{and} \quad \|\Lambda(\hat{f}(A\hat{x}) - \bar{f}(A\bar{x}))\|_p^p \leq F_0$$

Thus $(\hat{f}, \hat{x}) \in T_0$.

From now on suppose that $(\hat{f}, \hat{x}) \in T_i$ and we will argue that $(\hat{f}, \hat{x}) \in T_{i+1}$. We will continue to bound (22). Suppose that $\frac{KY_i}{X_i} \geq \epsilon$ for some $K \geq 1$. Then we can upper bound $R_i^\beta F_0^\theta$ as follows.

$$R_i^\beta F_0^\theta = \left(X_i\|\Lambda(\bar{f}(A\bar{x}) - b)\|_p^p + Y_i\|\Lambda A\bar{x}\|_p^p\right)^\beta \cdot C_2^\theta\left(\|\Lambda(\bar{f}(A\bar{x}) - b)\|_p^p + \epsilon\|\Lambda A\bar{x}\|_p^p\right)^\theta$$

$$\leq C_2^\theta\left(X_i\|\Lambda(\bar{f}(A\bar{x}) - b)\|_p^p + Y_i\|\Lambda A\bar{x}\|_p^p\right)^\beta\left(\|\Lambda(\bar{f}(A\bar{x}) - b)\|_p^p + \frac{KY_i}{X_i}\|\Lambda A\bar{x}\|_p^p\right)^\theta$$

$$\leq \left(\frac{C_2}{X_i}\right)^\theta\left(X_i\|\Lambda(\bar{f}(A\bar{x}) - b)\|_p^p + Y_i\|\Lambda A\bar{x}\|_p^p\right) \quad (\text{since } \beta + \theta = 1)$$

Thus, by the optimality of $\hat{f}, \hat{x}$,

$$\|\Lambda A\hat{x}\|_p^p \leq \frac{1}{\epsilon}\left(\|S\Lambda(\bar{f}(A\bar{x}) - b)\|_p^p - \|S\Lambda(\hat{f}(A\hat{x}) - b)\|_p^p\right) + \|\Lambda A\bar{x}\|_p^p$$

$$\leq \frac{1}{\epsilon}\left(\|\Lambda(\bar{f}(A\bar{x}) - b)\|_p^p - \|\Lambda(\hat{f}(A\hat{x}) - b)\|_p^p + C_3\epsilon R_i^\beta F_0^\theta\right) + \|\Lambda A\bar{x}\|_p^p \quad \text{(by (22))}$$

By our assumptions, it holds that

$$\|\Lambda(\bar{f}(A\bar{x}) - b)\|_p^p - \|\Lambda(\hat{f}(A\hat{x}) - b)\|_p^p \lesssim \epsilon(\|\Lambda(\bar{f}(A\bar{x}) - b)\|_p^p + \epsilon\|\Lambda A\bar{x}\|_p^p) \lesssim \epsilon R_i^\beta F_0^\theta.$$

It thus follows that

$$\|\Lambda A\hat{x}\|_p^p \leq C_4(R_i^\beta F_0^\theta) + \|\Lambda A\bar{x}\|_p^p$$

$$\leq C_4\left(\frac{C_2}{X_i}\right)^\theta\left(X_i\|\Lambda(\bar{f}(A\bar{x}) - b)\|_p^p + Y_i\|\Lambda A\bar{x}\|_p^p\right) + \|\Lambda A\bar{x}\|_p^p \quad (23)$$

$$\leq X_{i+1}\|\Lambda(\bar{f}(A\bar{x}) - b)\|_p^p + Y_{i+1}\|\Lambda A\bar{x}\|_p^p,$$

where

$$X_{i+1} = C_4 C_2^\theta X_i^{1-\theta}, \quad Y_{i+1} = 1 + \frac{C_4 C_2^\theta K Y_i}{X_i^\theta}.$$

Define $R_{i+1}$ to be the minimum of $R_0$ and the right-hand side of (23), i.e.

$$R_{i+1} = R_0 \wedge \left(X_{i+1}\|\Lambda(\bar{f}(A\bar{x}) - b)\|_p^p + Y_{i+1}\|\Lambda A\bar{x}\|_p^p\right).$$

We immediately have $(\hat{f}, \hat{x}) \in T_{i+1}$ which is needed to iterate the argument.

Let $X_0 = \frac{100}{\epsilon}$ and $Y_0 = 1$. By induction one can show that

$$X_i = C_5^{\frac{1-(1-\theta)^i}{\theta}}\left(\frac{100}{\epsilon}\right)^{(1-\theta)^i}$$

for $C_5 = C_4 C_2^\theta$. Then $C_6 \leq X_i \leq \frac{100}{\epsilon}$ for some constant $C_6$ for all $i \leq r$, thus $Y_{i+1} \leq 1 + C_7 Y_i \leq C_8 Y_i$ for some constants $C_7$ and $C_8$.

When $r \sim_p \ln\ln\frac{1}{\epsilon}$,

$$X_r \leq C_9 \quad \text{and} \quad Y_r \leq C_4(C_8)^{r-1} = \text{poly}\left(\ln\frac{1}{\epsilon}\right).$$

We shall also verify that $\frac{KY_i}{X_i} \geq \epsilon$ for small $\epsilon$. Indeed,

$$\frac{Y_i}{X_i} \geq \frac{1}{\frac{(100C_5^\theta)}{\epsilon}} \geq \frac{\epsilon}{K}$$

for $K = 100C_5^\theta$.

We iterate the above argument $r$ times. The total failure probability is at most $\delta r + 0.03 = 0.1$ since $\delta \sim 1/r$. It then follows from (22) with $i = r - 1$ that

$$\|\Lambda(\hat{f}(A\hat{x}) - b)\|_p^p - \|\Lambda(\bar{f}(A\bar{x}) - b)\|_p^p$$

$$\leq \|S\Lambda(\hat{f}(A\hat{x}) - b)\|_p^p - \|S\Lambda(\bar{f}(A\bar{x}) - b)\|_p^p + C_3\epsilon R_{r-1}^\beta F_0^\theta$$

$$\overset{(a)}{\leq} \epsilon\|\Lambda A\bar{x}\|_p^p + C_3\epsilon R_{r-1}^\beta F_0^\theta$$

$$\overset{(b)}{\lesssim} \epsilon\|\Lambda A\bar{x}\|_p^p + \epsilon\left(X_r\|\Lambda(\bar{f}(A\bar{x}) - b)\|_p^p + Y_r\|\Lambda A\bar{x}\|_p^p\right)$$

$$\lesssim \epsilon\|\Lambda(\bar{f}(A\bar{x}) - b)\|_p^p + \left(\epsilon\,\text{poly}\left(\ln\frac{1}{\epsilon}\right)\right)\|\Lambda A\bar{x}\|_p^p,$$

where step (a) is due to the optimality of $\hat{f}$ and $\hat{x}$ and step (b) to (23). Rescaling $\epsilon\,\text{poly}(\ln\frac{1}{\epsilon})$ to $\epsilon$ proves the claimed result, with additional $\text{poly}\log\frac{1}{\epsilon}$ factors in the lower bound for $\alpha$. $\square$

We are now ready to prove our main theorem of $(1 + \epsilon)$-approximation, Theorem 3.2.

**Theorem 3.2.** *Let $p \geq 1$ be a constant, $A \in \mathbb{R}^{n \times d}$, $b \in \mathbb{R}^n$, $\epsilon \in (0, 1)$ be sufficiently small and $\Lambda$ be an $n \times n$ diagonal matrix satisfying $\Lambda_{ii} > 0$ and $w_i(\Lambda A) \lesssim d/n$ for all $i$. There exists a randomized algorithm which, with probability at least $0.9$, makes $O(d^{1 \vee \frac{p}{2}}/\epsilon^{p \vee 2} \operatorname{poly} \log n)$ queries to the entries of $b$ and returns $(\hat{f}, \hat{x}) \in \mathsf{Lip}_1 \times \mathbb{R}^d$ satisfying the mixed error guarantee (3). The hidden constants in the bounds on the number of queries depends on $p$ only.*

*Proof.* Let $(\hat{f}, \hat{x}) = \arg\min_{x \in \mathbb{R}^d, f \in \mathsf{Lip}_1} \|S\Lambda(f(Ax) - b)\|_p^p + \epsilon\|\Lambda Ax\|_p^p$ and $\mathsf{OPT} = \|\Lambda(\bar{f}(A\bar{x}) - b)\|_p^p$. By the optimality of $(\bar{f}, \bar{x})$, we have

$$\|\Lambda(\bar{f}(A\bar{x}) - b)\|_p^p - \|\Lambda(\hat{f}(A\hat{x}) - b)\|_p^p \leq 0$$

By Theorem F.1, with probability at least $0.9$, it holds that

$$\left|\left(\|S\Lambda(\hat{f}(A\hat{x}) - b)\|_p^p - \|\Lambda(\hat{f}(A\hat{x}) - b)\|_p^p\right) - \left(\|S\Lambda(\bar{f}(A\bar{x}) - b)\|_p^p - \mathsf{OPT}\right)\right| \leq \epsilon(\mathsf{OPT} + \|\Lambda A\bar{x}\|_p^p).$$

This implies that

$$\begin{aligned}
\|\Lambda(\hat{f}(A\hat{x}) - b)\|_p^p - \mathsf{OPT} &\leq \|S\Lambda(\hat{f}(A\hat{x}) - b)\|_p^p - \|S\Lambda(\bar{f}(A\bar{x}) - b)\|_p^p + \epsilon(\mathsf{OPT} + \|\Lambda A\bar{x}\|_p^p) \\
&\leq \epsilon\|\Lambda A\bar{x}\|_p^p + \epsilon(\mathsf{OPT} + \|\Lambda A\bar{x}\|_p^p) \quad \text{(by the optimality of } \hat{f}, \hat{x}) \\
&\leq \epsilon\mathsf{OPT} + 2\epsilon\|\Lambda A\bar{x}\|_p^p.
\end{aligned}$$

Therefore,

$$\|\Lambda(\hat{f}(A\hat{x}) - b)\|_p^p \leq (1 + \epsilon)\mathsf{OPT} + 2\epsilon\|\Lambda A\bar{x}\|_p^p.$$

Rescaling $\epsilon$ completes the proof. $\qquad\square$

## G. The case $p = 1$

In this section, we shall first prove an analogous version of Lemma E.1 for $p = 1$. The issue is that in Lemma D.3, our entropy bound $\ln \mathcal{N}(\pi_1(T), d_T^\phi, \epsilon)$ grows too rapidly and is thus insufficient to control Dudley's integral

$$\int_0^\infty \sqrt{\ln \mathcal{N}(\pi_1(T), d_T^\phi, \epsilon)} d\epsilon,$$

since the resulting upper bound integral diverges. To address this issue, we instead invoke the more sophisticated control (8) of the supremum of a subgaussian process, which allows us to truncate the Dudley's integral at a suitably chosen $\epsilon_0 > 0$.

Specifically, we obtain that

$$\mathbb{E}_\xi \left(\sup_{(f,x) \in T} |\langle \xi_I, Z_I(f, x)\rangle - \langle \xi_I, Z_I(\bar{f}, \bar{x})\rangle|\right)^\ell \leq C^\ell(A + B + \sqrt{\ell} \operatorname{diam}(T, D_2))^\ell, \tag{24}$$

where

$$A := \left(\mathbb{E}_\xi \left(\sup_{\substack{(f_1,x_1),(f_2,x_2) \in T \\ D_2((f_1,x_1),(f_2,x_2)) \leq C(\alpha F)^{1/2}\epsilon_0}} |\langle \xi_I, Z_I(f_1, x_1) - Z_I(f_2, x_2)\rangle|\right)^\ell\right)^{\frac{1}{\ell}},$$

$$B := \int_{C(\alpha F)^{1/2}\epsilon_0}^\infty \sqrt{\ln \mathcal{N}(T, D_2, \epsilon)} d\epsilon.$$

Here, $C$ is the hidden constant in (14) and the truncation parameter $\epsilon_0$ is to be determined later.

By repeating the arguments in Lemma C.2, Lemma D.3 and Corollary D.4, we can bound the integral term $B$ as

$$B \lesssim \alpha\Gamma\sqrt{\ln \frac{n}{\epsilon d}} \left(\ln^{\frac{5}{4}} d + \ln \frac{\sqrt{M \ln \kappa}}{\epsilon_0}\right),$$

where $M = (d/n)R$, $\Gamma = (d/(\alpha n))^{1/2}F^{1/2}R^{1/2}$ and $\kappa = n/d$. Note that $\alpha\Gamma = (\alpha F)^{1/2}M^{1/2}$.

Next we upper bound $A$. Recall that

$$D_2((f_1, x_1), (f_2, x_2)) = \|Z_I(f_1, x_1) - Z_I(f_2, x_2)\|_2.$$

Applying Cauchy-Schwarz inequality yields

$$A \leq \left(\mathop{\mathbb{E}}_{\xi}\left(\sup_{\substack{(f_1,x_1),(f_2,x_2)\in T \\ D_2((f_1,x_1),(f_2,x_2))\leq C(\alpha F)^{1/2}\epsilon_0}} \|\xi_I\|_2\|Z_I(f_1, x_1) - Z_I(f_2, x_2)\|_2\right)^{\ell}\right)^{\frac{1}{\ell}} \leq C\sqrt{|I|}(\alpha F)^{\frac{1}{2}}\epsilon_0.$$

Combining (24) with (12) and (13), we obtain that

$$\left(\mathop{\mathbb{E}}_{S} \Psi_3^{\ell}\right)^{\frac{1}{\ell}} \lesssim \frac{1}{\alpha}\left(\mathop{\mathbb{E}}_{S}|I|^{\frac{\ell}{2}}\right)^{\frac{1}{\ell}}(\alpha F)^{\frac{1}{2}}\epsilon_0 + \Gamma\sqrt{\ln\frac{n}{\epsilon d}}\left(\ln^{\frac{5}{4}}d + \ln\frac{\sqrt{M\ln\kappa}}{\epsilon_0} + \sqrt{\ell}\right).$$

Choose $\epsilon_0$ such that

$$\sqrt{\alpha n}\epsilon_0(\alpha F)^{1/2} = \alpha\Gamma, \quad \text{that is,} \quad \epsilon_0 = \frac{\sqrt{M}}{\sqrt{\alpha n}}.$$

Since $|I| \sim \mathrm{Bin}(|J|, \alpha)$ and $|J| \leq n$, Lemma A.3 implies that

$$\mathbb{E}|I|^{\frac{\ell}{2}} \leq \left(\alpha|J| + \frac{\ell}{2}\right)^{\frac{\ell}{2}} \leq (\alpha n\ell)^{\frac{\ell}{2}}.$$

Substituting this bound yields

$$\left(\mathop{\mathbb{E}}_{S} \Psi_3^{\ell}\right)^{\frac{1}{\ell}} \lesssim \Gamma\sqrt{\ell} + \Gamma\sqrt{\ln\frac{n}{\epsilon d}}\left(\ln^{\frac{5}{4}}d + \ln(\alpha n\ln\kappa) + \sqrt{\ell}\right) \lesssim \Gamma\ln\sqrt{\frac{n}{\epsilon d}}\left(\ln^{\frac{5}{4}}d + \ln n + \sqrt{\ell}\right).$$

Finally, setting $\ell = \log(1/\delta)$ and applying Markov's inequality, we conclude that with probability at least $1 - \delta$,

$$\Phi_3 \lesssim \Gamma\sqrt{\ln\frac{n}{\epsilon d}}\left(\ln^{\frac{5}{4}}d + \ln n + \sqrt{\ln\frac{1}{\delta}}\right).$$

Thus, an analogous version of Lemma E.1 holds for $p = 1$ as well (with an additional $\log n$ term).

The same arguments establish corresponding versions of the remaining results in Appendix E, which includes Theorem 3.1. As a result, we obtain the $p = 1$ version of Theorem F.1, with the modification that the condition on $\alpha$ requires an additional $\log n$ factor.

## H. Missing Proofs for Non-adaptive Lower Bound

**Lemma 4.1.** *Let $\mathcal{A}$ be a randomized algorithm that takes* nonadaptive *samples of $b$: it fixes a set $S \subseteq [2Ns]$ of indices with $|S| = q$, reads $\{b_j : j \in S\}$ and outputs $(\hat{i}, \hat{\sigma})$. If $\Pr\{(\hat{i}, \hat{\sigma}) = (i^*, \sigma^*)\} \geq 2/3$, then $q \geq (2Ns)/3$.*

*Proof.* Let $E$ be the event that $J^* \in S$, i.e. the spike is sampled by the algorithm. Then

$$\Pr(E) = \frac{q}{2Ns}.$$

It is clear that when conditioned on $\neg E$, $b_S$ is independent of $\sigma^*$. Thus

$$\mathop{\Pr}_{\mathcal{A}}\{\hat{\sigma} = \sigma^* \mid \neg E\} = \mathop{\Pr}_{\mathcal{A}}\{\hat{\sigma} = -\sigma^* \mid \neg E\}$$

so

$$\Pr_{\mathcal{A}}\{\hat{\sigma} = \sigma^* \mid \neg E\} = \frac{1}{2}.$$

On the other hand, conditioned on $E$, the algorithm may succeed with probability at most $1$. Hence,

$$\Pr\{(\hat{i}, \hat{\sigma}) = (i^*, \sigma^*)\} \leq \Pr(E) \cdot 1 + \Pr(\neg E) \cdot \frac{1}{2} = \frac{1}{2} + \frac{\Pr(E)}{2} = \frac{1}{2} + \frac{q}{4Ns}.$$

Requiring this to be at least $2/3$ implies that $q \geq (2Ns)/3$, as claimed. $\qquad\square$

The remainder of this section is devoted to proving Theorem 4.2. First, we analyse the minimizer of $\|f(Ax) - b\|_p$ for our hard instance.

### Properties of Minimizer

For simplicity, let $\Phi(x) := \|f(Ax) - b\|_p^p$ and $x^*$ a minimizer of $\Phi(x)$. Define the two canonical candidates

$$x_+ := \epsilon a_{i^*}, \qquad x_- := -\epsilon a_{i^*}.$$

We slightly abuse the notation and write $x_{\sigma^*} \in \{x_+, x_-\}$ to denote the vector corresponding to the sign of $\sigma^*$. Also define

$$u^* := \max_x f(x) = \epsilon d - T\epsilon\sqrt{d}, \qquad \theta := \frac{T}{\sqrt{d}}.$$

**Lemma H.1.** *For all $i \neq i^*$ and for both $\tau \in \{\pm 1\}$, it holds that*

$$f(\langle \tau a_i, x_+\rangle) = 0 \qquad and \qquad f(\langle \tau a_i, x_-\rangle) = 0.$$

*Proof.* Fix $i \neq i^*$ and $\tau \in \{\pm 1\}$. Then

$$|\langle \tau a_i, x_+\rangle| = \epsilon|\langle a_i, a_{i^*}\rangle| \leq C\epsilon\sqrt{d} = T\epsilon\sqrt{d},$$

so $f(\langle \tau a_i, x_+\rangle) = 0$ by definition of $f$. The argument for $x_-$ is identical. $\qquad\square$

**Lemma H.2.** *The following statements hold.*

  (i)  $x_{\sigma^*}$ *is the minimizer;*
 (ii)  $\Phi(x_{\sigma^*}) = \left(1 + (1 - \epsilon^p)\theta^p + (1 + \epsilon\theta)^p\right)d^p$;
(iii)  $\Phi(x_{-\sigma^*}) - \Phi(x_{\sigma^*}) \geq \frac{p}{2}(1 - 2^{-(p-1)})\epsilon d^p$.

*Proof.* We shall prove (ii) and (iii) first.

**Proof of (ii)** By Lemma H.1, all non-planted blocks contribute $0$ to $\Phi(x_{\sigma^*})$. Next we examine the planted blocks.

- On the $\sigma^*$-block, we have $\langle \sigma^* a_{i^*}, x_{\sigma^*}\rangle = \epsilon d$ and $f(\langle \sigma^* a_{i^*}, x_{\sigma^*}\rangle) = u^*$. Thus, on the $s - 1$ non-spike coordinates (where $b = \epsilon d$), the residual is $\epsilon d - u^* = T\epsilon\sqrt{d}$, and on the spike coordinate (where $b = \epsilon d + d$), the residual is $d + \epsilon d - u^* = d + T\epsilon\sqrt{d}$.

- On the $(-\sigma^*)$-block, we have $\langle -\sigma^* a_{i^*}, x_{\sigma^*}\rangle = -\epsilon d \leq 0$ and so $f(\langle -\sigma^* a_{i^*}, x_{\sigma^*}\rangle) = 0$. Since $b = \epsilon d$ on all $s$ coordinates, residual is $\epsilon d$ on all $s$ coordinates.

Summing yields

$$\Phi(x_{\sigma^*}) = s(\epsilon d)^p + (s - 1)(T\epsilon\sqrt{d})^p + (d + T\epsilon\sqrt{d})^p = d^p + (s - 1)(T\epsilon\sqrt{d})^p + (d + T\epsilon\sqrt{d})^p,$$

which gives exactly (ii) after rearranging the terms.

**Proof of (iii)** Again, by Lemma H.1, all non-planted blocks contribute 0. The contribution to $\Phi(x_{-\sigma^*})$ must come from the pair of planted blocks and

$$\Phi(x_{-\sigma^*}) = s(\epsilon d - u^*)^p + (s-1)(\epsilon d)^p + (d + \epsilon d)^p$$

Therefore,

$$\begin{aligned}
\Phi(x_{-\sigma^*}) - \Phi(x_{\sigma^*}) &= ((d + \epsilon d)^p - (d + \epsilon d - u^*)^p) - ((\epsilon d)^p - (\epsilon d - u^*))^p \\
&= (d + \epsilon d)^p - (d + T\epsilon\sqrt{d})^p - ((\epsilon d)^p - (T\epsilon\sqrt{d}))^p \\
&= p((d + \xi_1)^{p-1} - \xi_2^{p-1})(\epsilon d - T\epsilon\sqrt{d}),
\end{aligned}$$

where the last line follows from the mean-value theorem and $\xi_1, \xi_2 \in [T\epsilon\sqrt{d}, \epsilon d]$. Note that

$$(d + \xi_1)^{p-1} - \xi_2^{p-1} \geq d^{p-1} - (\epsilon\theta)^{p-1}d^{p-1} \geq \left(1 - \frac{1}{2^{p-1}}\right)d^{p-1} \quad \text{and} \quad \epsilon d - T\epsilon\sqrt{d} \geq \frac{\epsilon d}{2},$$

we obtain that

$$\Phi(x_{-\sigma^*}) - \Phi(x_{\sigma^*}) \geq \frac{p}{2}\left(1 - \frac{1}{2^{p-1}}\right)\epsilon d^p.$$

**Proof of (i)** Now we show that $x_{\sigma^*}$ is the minimizer. Fix $x$ and let $\sigma \in \{-1, 1\}$ be the sign such that $t = \langle \sigma a_{i^*}, x \rangle > 0$. Consider the pair of planted blocks.

- If $\sigma = \sigma^*$, then the $(\sigma^*)$-block contributes to $\Phi(b)$ at least $(s-1)(\epsilon d - f(t))^p + (d + \epsilon d - f(t))^p$ and the $(-\sigma^*)$-block contributes to $\Phi(b)$ exactly $s(\epsilon d)^p$. Hence

$$\Phi(x) \geq s(\epsilon d)^p + (s-1)(\epsilon d - f(t))^p + (d + \epsilon d - f(t))^p \geq \Phi(x_{\sigma^*}),$$

  where the second inequality follows from the fact that $f(t) \leq u^*$ and the middle expression is minimized when $t = u^*$.

- If $\sigma = -\sigma^*$, then the $(\sigma^*)$-block contributes to $\Phi(b)$ at least $(s-1)(\epsilon d)^p + (d + \epsilon d)$ and the $(-\sigma^*)$-block contributes at least $s(\epsilon d - f(t))^p$. Hence

$$\Phi(x) \geq s(\epsilon d - f(t))^p + (s-1)(\epsilon d)^p + (d + \epsilon d)^p \geq \Phi(x_{-\sigma^*}), \tag{25}$$

  where the last inequality follows from the fact that the middle expression is minimized when $f(t) = u^*$.

Therefore, it always holds that $\Phi(x) \geq \Phi(x_{\sigma^*})$ and so $x_{\sigma^*}$ is the minimizer. $\qquad\square$

**Lemma H.3.** *It holds that*

$$\|Ax_{\sigma^*}\|_p^p \leq 2(C^p + 1)d^p.$$

*Proof.* It holds for $i \neq i^*$ that

$$|\langle a_i, x_{\sigma^*}\rangle| \leq \epsilon C\sqrt{d}, \qquad |\langle -a_i, x_{\sigma^*}\rangle| \leq \epsilon C\sqrt{d}.$$

For $i = i^*$, it holds that

$$|\langle a_{i^*}, x_{\sigma^*}\rangle| = \epsilon d, \qquad |\langle -a_{i^*}, x_{\sigma^*}\rangle| = \epsilon d.$$

Therefore

$$\|Ax_{\sigma^*}\|_p^p \leq 2s(N-1)(C\epsilon\sqrt{d})^p + 2s(\epsilon d)^p \leq 2C^p d^p + 2d^p \leq 2(C^p + 1)d^p. \qquad\square$$

**Identification of $i^*$ and $\sigma^*$**

Now we are ready to prove Theorem 4.2.

**Theorem 4.2.** *Let $K = 4^p(C^p + 1)$. Suppose that $\hat{x}$ satisfies*

$$\|f(A\hat{x}) - b\|_p^p \leq \left(1 + \frac{\epsilon}{K}\right)\|f(Ax^*) - b\|_p^p + \frac{\epsilon}{K}\|Ax^*\|_p^p,$$

*then it holds that*

$$i^* = \arg\max_i |\langle a_i, \hat{x}\rangle| \quad and \quad \mathrm{sgn}\langle a_{i^*}, \hat{x}\rangle = \sigma^*.$$

*Proof.* From the approximation guarantee,

$$E := \|f(A\hat{x}) - b\|_p^p - \|f(Ax^*) - b\|_p^p \leq \frac{\epsilon}{K}\left(\|f(Ax^*) - b\|_p^p + \|Ax^*\|_p^p\right).$$

By the Lemma H.2(i)(ii), we have $\|f(Ax^*) - b\|_p^p \leq C_0 d^p$ for $C_0 = 2^p + 2$ and $\|Ax^*\|_p^p \leq C_1 d^p$ for $C_1 = 2(C^p + 1)$. Hence

$$E \leq \frac{\epsilon}{K}(C_0 + C_1)d^p =: \epsilon\eta d^p.$$

Define

$$u_i^+ := f(\langle a_i, \hat{x}\rangle), \qquad u_i^- := f(\langle -a_i, \hat{x}\rangle), \qquad u_i := \max\{u_i^+, u_i^-\}.$$

$u_{i^*}$ **is large.** The contribution from the spike coordinate is at least

$$(\epsilon d + d - u_{i^*})^p \geq (\epsilon d + d - u^*)^p + pd^{p-1}(u^* - u_{i^*}),$$

where the inequality follows from the mean-value theorem. It follows that

$$u^* - u_{i^*} \leq \frac{E}{pd^{p-1}} \leq \frac{\epsilon\eta}{p}d,$$

or

$$u_{i^*} \geq u^* - \frac{\epsilon\eta}{p}d = \epsilon d\left(1 - \frac{\eta}{p} - \theta\right).$$

$u_i$ $(i \neq i^*)$ **is small.** For any $i \neq i^*$, both $(\pm a_i)$-blocks have $b = 0$ everywhere, so the contribution to $\|f(A\hat{x}) - b\|_p^p$ from the pair of blocks $\pm a_i$ is

$$s(u_i^+)^p + s(u_i^-)^p \geq su_i^p.$$

Since such blocks contribute $0$ to the optimum, we have

$$\sum_{i \neq i^*} su_i^p \leq E,$$

and therefore for each $i \neq i^*$,

$$u_i \leq \left(\frac{E}{s}\right)^{1/p} = \epsilon E^{1/p} \leq \epsilon d \cdot (\epsilon\eta)^{1/p}.$$

**Comparing $u_i$ and $u_{i^*}$.** We claim that $u_i \leq u_{i^*}$ for $i \neq i^*$. It suffices to show that

$$(\epsilon\eta)^{1/p} \leq 1 - \frac{\eta}{p} - \theta.$$

By our choice of $K$ and $d$, $\eta^{1/p} \leq 1/4$ and $\theta \leq 1/2$. The inequality above holds.

**Sign recovery**  We claim that $\sigma^* = \text{sgn}\langle a_{i^*}, \hat{x} \rangle$. Otherwise, on the $(\sigma^*)$-block, one would have $f(\langle a_{i^*}, \hat{x} \rangle) = 0$. By considering the pair of planted blocks, it follows from (25) that

$$\Phi(\hat{x}) \geq \Phi(x_{-\sigma^*}).$$

By Lemma H.2(iii),

$$E = \Phi(\hat{x}) - \Phi(x_{\sigma^*}) \geq \Phi(x_{-\sigma^*}) - \Phi(x_{\sigma^*}) \geq \frac{p}{2}\left(1 - \frac{1}{2^{p-1}}\right)\epsilon d^p.$$

This contradicts $E \leq \epsilon \eta d^p$ for $\eta \leq 1/4$. Therefore, it must hold that $\sigma^* = \text{sgn}\langle a_{i^*}, \hat{x} \rangle$. $\qquad\square$

