# OpenReview forum: "Active Regression for Single-Index Models with Unknown Link Functions"
_ICML.cc/2026/Conference — ICML 2026 regular_

### Official Review · Reviewer_m63L · 2026-03-08

**Soundness:** 4
**Presentation:** 4
**Significance:** 3
**Originality:** 3
**Overall Recommendation:** 5
**Confidence:** 4

**Summary:**

This paper provides upper and lower bounds about the problem of $\ell_p$ regression of a single index model with unknown Lipschitz link function in the active learning setting. Their upper bound a weighted sampling approach that has been used previously in the context of learning SIMs with known link function. The improvement that generalizes the result to this case is the use of a cover over Lipschitz functions.  Their lower bound improves the the prior lower bound that was not tight for $p>2$.

**Compliance With Llm Reviewing Policy:**

Affirmed.

**Final Justification:**

I changed my evaluation to acceptance. I scored the work as follows:

Strengths:
1) Soundness: The paper proves all assertions and it performs experiments.
2) Significance: The paper resolves a fundamental problem considered my many researchers.
3) Clarity: The presentation of the paper is very good and the results are very clear.
4) Originality: The lower bound has an original solution that escaped prior work.

Weaknesses:
1) Originality: The upper bound applies the technique from Li and Tai (2025) with a cover for all lipschitz functions to work for unknown f.

The rebuttal addressed my concerns and changed my evaluation.

**Key Questions For Authors:**

1) The algorithm relies on solving a sketched version of the problem and as noted by the authors there has been a lot of research in learning SIMs. However, algorithm for learning SIMs often have assumptions on the underlying function and the distribution. Could the sketched problem be harder to solve computationally than the original?
2) Could you comment a little bit more on the differences of the upper bound with the approach of Li & Tai, 2025 and the technical difficulties to extend it here?

**Limitations:**

Yes

**Strengths And Weaknesses:**

Strengths:
1) Presentation: The paper seems well-written and easy to follow.
2) Soundness: The paper supports all the claims with rigorous proofs as well as with experiments.
3) Significance: The paper handles a the task of SIM regression that is very popular in modern day theoretical research.
4) Originality: It seems that the lower bound question had been considered by previous work and it could not be easily attained.

Weaknesses:
1) It seems that the upper bound technique is very well known and used for previously for SIM regression.

---

> ### Author Rebuttal · Authors · 2026-03-28
>
> We thank the reviewer for the positive assessment and helpful questions.
>
> 1. We agree that solving the sketched problem can be computationally challenging, especially due to the non-convexity of SIM regression. This is already present in the original (unsketched) problem, and the sketch does not fundamentally change this aspect; rather, it reduces the problem scale.
>
>     Our focus is on query complexity. For the regression solver, theoretical guarantees are mainly known for isotron-type methods under monotonicity assumptions for $p=2$, while extensions to more general settings have largely been heuristic in the literature. Thus, while the sketch improves sample efficiency, theoretical guarantees of computational tractability remain open.
>
> 2. We agree that the high-level sampling framework (Lewis-weight-based sketching) follows prior work. The main technical difference from (Li \& Tai, 2025) is that we must handle an unknown $f$, which introduces a supremum over a joint pair $(x,f)$ that does not necessarily lie in a rectangle.
>
>     In spite of this, our analysis shows that it can be decoupled into handling fixed $f$ and fixed $x$. The case of fixed $x$ is new and requires suitable covering bounds over $f$. Lemma 1.3 provides such a bound in a weighted $\ell_\infty$ norm compatible with the structure of $f(Ax)$. This avoids specific net constructions (as in the COLT paper for $p=2$) and leads to a more modular argument. This is the main new ingredient beyond (Li \& Tai, 2025). See also our response to Reviewer TgJf.

---

> > ### Author Rebuttal · Reviewer_m63L · 2026-04-01
> >
> > The authors have answered my questions fully. I will consider to adjust score.

---

### Official Review · Reviewer_TgJf · 2026-03-09

**Soundness:** 3
**Presentation:** 3
**Significance:** 3
**Originality:** 2
**Overall Recommendation:** 3
**Confidence:** 3

**Summary:**

The paper considers active regression for single-index model with unknown Lipschitz link function. Using the same algorithm as in Li & Tai (2025) together with the minimization over the link function f, the same theoretical guarantee in Li & Tai (2025) for known link is proved to still hold. The main technical piece is a metric entropy bound for Lipschitz functions in weighted infinity norm. The paper also proves a matching query lower bound with known link function for p>2, and thus closed a gap left by Li & Tai (2025).

**Compliance With Llm Reviewing Policy:**

Affirmed.

**Final Justification:**

Since the authors' rebuttal is coherent with my original understanding of the paper, I decide to keep my original score.

**Key Questions For Authors:**

I do not have any question. In terms of active regression for single-index model with known and unknown link functions, the paper, together with Li & Tai (2025), provide very complete answers. Apart from standard questions regarding logarithmic factors, more important questions such as adaptive sampling/queries seem to be out of the scope of the paper. The paper can probably be rejected or accepted as is.

**Limitations:**

Discussed above.

**Strengths And Weaknesses:**

The paper is well written in terms of main contributions and main technical novelties compared with the results of the literature. As someone who knows nothing about the area, it was a pleasant read with sufficient background information. My overall impression is that the paper is an immediate follow-up or extension of the work of Li & Tai (2025). While the results are technically solid, it is unclear whether the actual contribution is significant or just incremental. My subjective assessment is weak reject (though I do not know the area, I did read the related work of Li & Tai (2025) to help me understand some of the technical proofs). I would be happy to raise my score if other expert reviewers have different opinions.

Comments on the upper bound. After reading Li & Tai (2025), it is clear that the main challenge of extension to the case with unknown link function is not algorithmic design, but only the theoretical analysis. While I appreciate that the authors are very clear that their main technical contribution is the metric entropy bound on Lipschitz functions, I do not find it especially novel or technically involved. The vast literature on metric entropy and empirical process theory has already dealt with unbounded support by localizing them on bounded pieces and controlling tails/weights and then pasting together bounded-domain pieces. The statement of Lemma 1.3 is certainly new (to the best of my knowledge), but similar ideas of the proof have already been used in the literature for the analyses of similar quantities. Apart of Lemma 1.3, there is certainly still much to be done given that the supremum is not taken over a rectangle. However, the majority of the proof structure is already done and available in Li & Tai (2025).

Comments on lower bound. Here I do think the authors have solved an open problem that was left by Li & Tai (2025) for p>2. The lower bound for p>2 given by Li & Tai (2025) is of order d/\epsilon^p, if I understand correctly. In Li & Tai (2025), the lower bound construction involves d independent blocks, each drawn from D_0 or D_1. The reduction shows that any good approximate solution will reveal at least 2d/3 blocks, thus leading to a lower bound linear in d. The current paper has a very different approach. Its lower bound construction is a packing argument of cardinality d^{p/2}, and each is repeated \epsilon^{-p} times. Any good approximate solution is shown to correlate with the optimal solution. To be honest, the current construction is quite standard in the lower bound construction of many other high-dimensional recovery problems. Instead of saying that there is a huge improvement over Li & Tai (2025), I feel it is more accurate to say that the lower bound construction in Li & Tai (2025) is highly sub-optimal. A more experienced researcher would certainly do the packing strategy instead of the one in Li & Tai (2025). Nonetheless, this is still a valid contribution that nicely closed an open problem.

---

> ### Author Rebuttal · Authors · 2026-03-28
>
> We thank the reviewer for the careful reading and for recognizing that the paper, together with prior work, gives a nearly complete picture of active regression for single-index models. We clarify the novelty below.
>
> 1. A key baseline is the COLT paper of Gajjar et al., which handled $p=2$ and introduced the unknown $f$ setting. While Li \& Tai (2025) moved toward general $p$, it was limited to the "known $f$" case. Our work addresses the remaining case of unknown $f$ and bridging this gap is a primary contribution.
>
>     Lemma 1.3 plays a central role: it yields a conceptually simpler and more structured analysis compared to prior work, whose arguments are specific to the $p=2$ geometry (and $(x,f)$ in a rectangle). Our approach isolates a core difficulty in the case of unknown $f$ into a single abstraction, which, combined with the framework in (Li \& Tai, 2025), applies uniformly to all $p\geq 2$. This not only significantly simplifies the proof but also conceptually explains why the case of unknown $f$ is do-able in a similar manner.
>
> 2. Regarding the lower bound, despite the common technique of packing, the weaker bound in prior work suggests our current construction is not immediate. We also think correcting sub-optimality in the literature is a core function of theoretical research.

---

> > ### Author Rebuttal · Reviewer_TgJf · 2026-04-03
> >
> > I thank the authors for emphasizing the contributions. I think these points are already clear in the original manuscript, and I do not have any misunderstandings regarding the results or the proofs. I therefore decide to keep the original score.

---

### Official Review · Reviewer_HDe8 · 2026-03-09

**Soundness:** 3
**Presentation:** 4
**Significance:** 2
**Originality:** 3
**Overall Recommendation:** 5
**Confidence:** 3

**Summary:**

The authors study the active regression problem with unknown link function $f$. The aim is $min_{f, x} \lVert f(Ax) - b \rVert_p^p$ where $f$ is a $1$-Lipschitz link function. In the active version of this problem, we are given full access to $A$, but have to query for entries of $b$. The goal is to query as few entries as possible and obtain a $(1 + \epsilon)$-approximate solution with constant probability.

While the setting with no or known link functions has received significant attention, only Gajjar et al. (2024) previously studied the setting of unknown link functions. They restricted their attention to $p = 2$, and obtained a query complexity of $O(d/(\epsilon^2 \log^{O(1)} n))$. In this article, the authors achieve query complexity $O(d^{max(1, p/2)} / \epsilon^{max(p, 2)} \log^{O(1)} n$. Notice that this matches the previous result for $p = 2$, but extends to all $p \geq 1$. They also show that for non adaptive queries and p > 2, this bound is tight up to poly-logarithmic factors.

**Compliance With Llm Reviewing Policy:**

Affirmed.

**Final Justification:**

I continue to believe that this paper is a good fit for ICML, and the obtained results are clearly above the acceptance bar.

**Key Questions For Authors:**

Could you add some extra motivation for unknown link functions? What are natural circumstances in which these arise?

**Limitations:**

yes

**Strengths And Weaknesses:**

Soundness: While I did not check all the claims in detail, the overall mathematical structure is sound. I do not believe that there are serious issues. The supporting experiments are thorough enough for a theory paper.

Presentation: The presentation of this article is very good. The main theoretical claims are clearly outlined, and informal theorems are used to allow the reader to develop an intuition before getting into the details that are harder to understand. Given the relatively heavy nature of this article, the authors manage to make it digestible to the average reader of ICML articles.

Significance: The setting is quite theoretical an d bit contrived, since it is relatively natural to assume to know the link function beforehand, especially if one was to extend this research to multi layer networks.

Originality: The algorithm builds on an established line of work, and resolves a natural open question. The algorithm is rather simple, but the analysis requires technical expertise/insights.

---

> ### Author Rebuttal · Authors · 2026-03-28
>
> We thank the reviewer for the positive assessment.
>
> The setting with unknown $f$ was introduced in this context by Gajjar et al. (COLT 2024), which we cited. As noted in that work, it is well-motivated in computational science, where the link function is often modeled as an unknown piecewise polynomial or piecewise constant function. More broadly, single-index models with unknown link functions are standard in nonparametric regression, where $f$ captures an unknown nonlinear transformation.
>
> From a theoretical perspective, the case of unknown $f$ is a strictly more general formulation that subsumes the case of known $f$. Our results show that this additional generality can be handled without increasing query complexity, which we view as a conceptual contribution.

---

> > ### Author Rebuttal · Reviewer_HDe8 · 2026-04-01
> >
> > The authors answered my questions.

---

### Official Review · Reviewer_HsTE · 2026-03-13

**Soundness:** 3
**Presentation:** 2
**Significance:** 2
**Originality:** 2
**Overall Recommendation:** 3
**Confidence:** 2

**Summary:**

This paper studies active regression for single-index models with unknown link function and oracle query access to labels. The authors consider $\ell_p$ loss for regression, and extend the prior algorithm and guarantees for $p=2$ under unknown link function to general $p \geq 1$. They prove an upper bound of $\tilde{O}(d^{p/2 \vee 1}/\epsilon^{p \vee 2})$ on the number of queries to achieve a $1 + \epsilon$ multiplicative approximation error caused by sampling. When $p \geq 2$, they prove a nearly matching lower bound of $\Omega(d^{p/2 \vee 1}/\epsilon^{p \vee 2})$.

**Compliance With Llm Reviewing Policy:**

Affirmed.

**Final Justification:**

I thank the authors for their detailed response. However, my main concern about novelty still remains. After reading their response, I'm not sure if the contributions provide enough novel intuitions for the ICML community, but I'm keeping my confidence score at 2 since I'm not very familiar with the prior work.

**Key Questions For Authors:**

1. I’m struggling to understand the number of queries to the coordinates of $b$ in Algorithm 2. $k_i$ is defined as $\lceil n w_i / d\rceil$, therefore $k_i \geq 1$ whenever $w_i > 0$. Whenever $k_i \geq 1$, we need to make a query for that particular coordinate in $b$. Does this mean that at most $\tilde{O}(d^{1\vee p/2}/\epsilon^{p\vee 2})$ of $w_i$s are not zero? In any case, $k_i = 0$ or $k_i =1$ would be very sensitive to the computation of $w_i$, which doesn’t seem reasoanble. Perhaps the definition of $k_i$ should be modified.

2. I think it would be helpful if the authors could mention the general complexity of obtaining Lewis weights (line 1) and solving the regression problem (line 6) in Algorithm 2.

3. Is Lemma 2.2 a contribution of this work? There is no reference to it in the main text and no particular intuition about its importance and connection to the main results.

4. Why does Theorem 4.3 need the algorithm to be deterministic?

5. What was the motivation behind using the Communities dataset for the experiment? Given the theoretical nature of the work, synthetic experiments with finer control on $f$ and data could provide more insights.

**Limitations:**

I think the authors can highlight limitations of practical implementations of Algorithm 2 better, e.g. the need to solve the non-convex minimization problem. This is not a particular limitation of this work, but discussing it would better present the scope of this work.

**Strengths And Weaknesses:**

*Strengths*: The proof techniques and intuitions are clearly discussed in the main text, and the authors' characterization of the query complexity is sharp via a matching lower bound.

*Weaknesses*: I am not familiar with the literature discussed in this paper on active regression of single-index models. To me, it seems like most of the analysis follows from Li and Tai, 2025, and I'm not sure if there's enough technical novelty in this work. The authors mention Lemma 1.3 as a core technical contribution, but its derivation seems somewhat standard. Also, the conceptual difference between the lower bound argument in this paper and in previous papers is not entirely clear to me.

Additionally, there are certain ambiguities in the text that make understanding a bit challenging. These are discussed in the Questions section.

---

> ### Author Rebuttal · Authors · 2026-03-28
>
> We thank the reviewer for the careful reading and helpful suggestions. We refer to our response to Reviewer TgJf for clarification on novelty. We address the specific questions below.
>
> 1. The Lewis weight $w_i = 0$ only if the row is a zero row; but the $i$-th row is sampled via a binomial variable $N(k_i,\\alpha)$ with small $\\alpha\\approx d^{p/2}/(\\epsilon^p n)$ (Line 4 of Algorithm 1). Thus even if $k_i\\geq 1$, the row may not be selected. Thus, the sampling is not overly sensitive to $k_i$. It is clear that the expected number of sampled rows is $O(\alpha n)$, and and by standard concentration bounds, it remains $O(\alpha n)$ with high probability.
>
> 2. (i) We would like to point out that exact Lewis weights are unnecessary; constant-factor approximations suffice (affecting sample complexity by only a constant factor). For $p<4$, they can be computed in $O(\log \log n)$ rounds of leverage score estimation [CP15], each taking $\tilde{O}(nd + \operatorname{poly}(d))$ time. For $p\geq 4$, they can be computed in $\tilde{O}(d)$ rounds of leverage score estimation [AGS24]. The overall runtimes are $\tilde{O}(nd+\operatorname{poly}(d))$ and $\tilde{O}(nd^2 + \operatorname{poly}(d))$, respectively. We will include this in the next version of the paper.
>
>     (ii) Our focus is query complexity. For the regression solver, theoretical complexity guarantees are known for isotron-type methods mainly for $p=2$ and monotone functions $f$ (without assuming convexity). Extensions to other cases have largely been heuristic in literature.
>
>     Assuming increasing $f$, convergence to critical points for $p>2$ can be established but obtaining explicit convergence rate remains challenging. These issues are beyond the scope of our current work and so we did not include them.
>
> [CP15] Michael B. Cohen and Richard Peng. 2015. $L_p$ Row Sampling by Lewis Weights. In Proceedings of STOC 2015.
>
> [AGS24] Simon Apers, Sander Gribling and Aaron Sidford. On computing approximate Lewis weights. https://arxiv.org/abs/2404.02881
>
> 3. Lemma 2.2 is a standard property of Lewis weights (also used in (Li \& Tai, 2025)) and not a contribution. We will add the citation. The lemma is used in the analysis and can be moved to the appendix.
>
> 4. This is after applying Yao's minimax principle (mentioned at the beginning of Section 4): proving a lower bound for deterministic algorithms under a randomized input distribution implies the same bound for randomized algorithms.
>
> 5. Our primary contribution is theoretical. We used the Communities and Crime dataset as a standard benchmark in prior work on single-index models; e.g., (Kakade et al, 2011), which studied isotron regression for $p=2$. We agree synthetic experiments would be valuable and will include them in the next version. In particular, we have conducted the following experiment.
>
> We consider ReLU link functions $f(x) = \alpha \max(0,x - \tau),$ with $\alpha \in [0,1]$. The true parameters are $\alpha^* = 0.8$ and $\tau^* = 0.5$. For the data generation, we set $n = 2000$, $d = 20$ and fix $x^\ast$ to be a unit vector. The rows $a_i$ of the matrix $A \in \mathbb{R}^{n \times d}$ are independent and identically distributed from the following mixture distribution
> $$
> a_i \sim
> \begin{cases}
>     N(8 x^\ast, 100 \Sigma) & \text{with probability} \quad 0.05 \\\\
>     N(0,\Sigma) & \text{with probability} \quad 0.95
> \end{cases}
> $$
> where the covariance matrix $\Sigma \in \mathbb{R}^{d \times d}$ is diagonal, i.e. $\Sigma = \operatorname{diag}(\sigma_1^2,\dots,\sigma_d^2)$ with $\sigma_j^2 = 1-\frac{j-1}{d-1}(0.9)$.
> The vector $b$ is defined as $b_i = f(a_i^T x^\ast) + \epsilon_i$, where $\epsilon_i$ are i.i.d. $N(0,0.05^2)$ variables.
>
> We then solve the optimization problem $\min_{x \in \mathbb{R}^d, f \in \mathcal{F}} \sum_i |{f(Ax)_i - b_i}|^p$ for $p \in \\{1,3 \\}$, where $\mathcal{F} = \{x \mapsto \alpha \max(0,x - \tau): \alpha \in [0,1], \tau \in \mathbb{R} \}$ is a ReLU family.
>
> We observe that Lewis-weight sampling consistently achieves lower error than uniform sampling at the same query budget. We report the median $\epsilon$ (approximation error) across sampling fractions 0.1, 0.15, ..., 0.3.
>
> For $p = 1$:
> - Uniform: $0.1793, 0.1953, 0.1372, 0.1154, 0.1445$,
> - Lewis: $0.1235, 0.1064, 0.1178, 0.1040, 0.1089$.
>
> For $p = 3$:
> - Uniform: $2.92 \times 10^{-4}, 2.44 \times 10^{-4}, 1.17 \times 10^{-4}, 1.89 \times 10^{-4}, 9.73 \times 10^{-5}$,
> - Lewis: $1.02 \times 10^{-5}, 8.13 \times 10^{-6}, 9.81 \times 10^{-6}, 5.24 \times 10^{-6}, 5.26 \times 10^{-6}$.

---

> > ### Author Rebuttal · Reviewer_HsTE · 2026-04-03
> >
> > Thank you for the detailed response and for the additional experiment. However, my main concern about novelty still remains. After reading their response, I'm not sure if the contributions provide enough novel intuitions for the ICML community, but I'm keeping my confidence score at 2 since I'm not very familiar with the prior work.

---

### Decision · Program_Chairs · 2026-04-30

**Decision:**

Accept (regular)

**Comment:**

This paper studies active regression for single-index models with an unknown Lipschitz link function in a query-efficient setting. The paper presents matching upper and lower bounds extending prior work, and the theoretical results are supplemented with empirical evaluations. Reviewers note the problem setting is well-motivated, the results are technically correct, and several non-trivial technical contributions beyond Li & Tai (2025), particularly in handling the unknown link function and tightening the lower bound for $p>2$. The main suggested improvements are to clarify parts of the presentation and to provide additional intuition for certain technical steps to improve readability in the final version.

Overall, the paper provides a solid technical contribution and resolves a natural extension of prior work, and is recommended for acceptance at ICML.